# Representation of visual landmarks in retrosplenial cortex

Lukas F Fischer, Raul Mojica Soto-Albors, Friederike Buck, Mark T Harnett*

Department of Brain and Cognitive Sciences, MGovern Institute for Brain Research, Massachusetts Institute of Technology, Cambridge, United States

**Abstract** The process by which visual information is incorporated into the brain's spatial framework to represent landmarks is poorly understood. Studies in humans and rodents suggest that retrosplenial cortex (RSC) plays a key role in these computations. We developed an RSC-dependent behavioral task in which head-fixed mice learned the spatial relationship between visual landmark cues and hidden reward locations. Two-photon imaging revealed that these cues served as dominant reference points for most task-active neurons and anchored the spatial code in RSC. This encoding was more robust after task acquisition. Decoupling the virtual environment from mouse behavior degraded spatial representations and provided evidence that supralinear integration of visual and motor inputs contributes to landmark encoding. V1 axons recorded in RSC were less modulated by task engagement but showed surprisingly similar spatial tuning. Our data indicate that landmark representations in RSC are the result of local integration of visual, motor, and spatial information.

## Introduction

Spatial navigation requires the constant integration of sensory information, motor feedback, and prior knowledge of the environment (*Hardcastle et al., 2015*; *McNaughton et al., 2006*; *Taube, 2007*; *Valerio and Taube, 2012*). Visual landmarks are particularly advantageous for efficient navigation, representing information-rich reference points for self-location and route planning (*Etienne et al., 1996*; *Gothard et al., 1996b*; *Jeffery, 1998*; *Knierim et al., 1995*; *McNaughton et al., 1991*). Even in situations where the immediate surroundings may not be informative, distal landmarks can provide critical orientation cues to find goal locations (*Morris, 1981*; *Tolman, 1948*). Their importance is further underlined by the fact that salient visuo-spatial cues anchor almost every type of spatially-tuned cells observed in the mammalian brain to date, including head-direction cells (*Jacob et al., 2017*; *Taube et al., 1990*; *Yoder et al., 2011*), hippocampal place cells (*Buzsáki, 2005*; *Jeffery, 1998*), and grid cells in the medial entorhinal cortex (*Hafting et al., 2005*; *Pérez-Escobar et al., 2016*). Even in scenarios where self-motion feedback is in conflict with external visual cues, landmarks exert powerful influence on the head-direction system (*Etienne and Jeffery, 2004*; *Valerio and Taube, 2012*). A number of theoretical studies have shown the importance of landmarks for error correction during spatial computations (*Burgess et al., 2007*; *Fuhs and Touretzky, 2006*; *Monaco et al., 2011*; *Sreenivasan and Fiete, 2011*). However, it remains poorly understood how visual information is integrated into spatial code for goal-directed behavior.

Converging evidence points to the retrosplenial cortex (RSC) as an important locus for landmark computations. Studies in humans with damage to RSC, as well as functional imaging studies, suggest a key role for RSC in utilizing familiar visual cues for navigation (*Auger et al., 2012*; *Epstein, 2008*; *Epstein and Vass, 2014*; *Maguire, 2001*; *Vann et al., 2009*). Additionally, RSC exhibits some of the earliest measurable metabolic dysfunction in Alzheimer's disease (AD) (*Minoshima et al., 1997*; *Pengas et al., 2010*; *Villain et al., 2008*). This is consistent with the putative roles of RSC in general mnemonic processing (*Cooper et al., 2001*; *Spiers and Maguire, 2006*; *Svoboda et al., 2006*;

*For correspondence:
harnett@mit.edu

Competing interests: The authors declare that no competing interests exist.

**eLife digest** When moving through a city, people often use notable or familiar landmarks to help them navigate. Landmarks provide us with information about where we are and where we need to go next. But despite the ease with which we – and most other animals – use landmarks to find our way around, it remains unclear exactly how the brain makes this possible.

One area that seems to have a key role is the retrosplenial cortex, which is located deep within the back of the brain in humans. This area becomes more active when animals use visual landmarks to navigate. It is also one of the first brain regions to be affected in Alzheimer's disease, which may help to explain why patients with this condition can become lost and disoriented, even in places they have been many times before.

To find out how the retrosplenial cortex supports navigation, Fischer et al. measured its activity in mice exploring a virtual reality world. The mice ran through simulated corridors in which visual landmarks indicated where hidden rewards could be found. The activity of most neurons in the retrosplenial cortex was most strongly influenced by the mouse's position relative to the landmark; for example, some neurons were always active 10 centimeters after the landmark.

In other experiments, when the landmarks were present but no longer indicated the location of a reward, the same neurons were much less active. Fischer et al. also measured the activity of the neurons when the mice were running with nothing shown on the virtual reality, and when they saw a landmark but did not run. Notably, the activity seen when the mice were using the landmarks to find rewards was greater than the sum of that recorded when the mice were just running or just seeing the landmark without a reward, making the "landmark response" an example of so-called supralinear processing.

Fischer et al. showed that visual centers of the brain send information about landmarks to retrosplenial cortex. But only the latter adjusts its activity depending on whether the mouse is using that landmark to navigate. These findings provide the first evidence for a "landmark code" at the level of neurons and lay the foundations for studying impaired navigation in patients with Alzheimer's disease. By showing that retrosplenial cortex neurons combine different types of input in a supralinear fashion, the results also point to general principles for how neurons in the brain perform complex calculations.

*Valenstein et al., 1987*) and route-planning (*Spiers and Maguire, 2006*), both of which are hallmarks of cognitive decline in AD patients (*Vann et al., 2009*). Lesion studies in rodents indicate that RSC is also important for navigating based on self-motion cues alone (*Elduayen and Save, 2014*). These findings are congruent with known RSC anatomy: situated at the intersection of areas that encode visual information, motor feedback, higher-order decision making, and the hippocampal formation (*van Groen and Wyss, 1992*; *Kononenko and Witter, 2012*; *Miyashita and Rockland, 2007*; *Sugar et al., 2011*), RSC is ideally positioned to integrate these inputs to guide ongoing behavior. Electrophysiological recordings in freely moving rats have shown that individual RSC neurons conjunctively encode space in egocentric and allocentric spatial reference frames (*Alexander and Nitz, 2015*). When placed in a one-dimensional environment, RSC neurons exhibit single, spatially tuned receptive fields (*Mao et al., 2017*), while in two-dimensional environments (*Alexander and Nitz, 2017*) they were found to express multiple receptive fields. RSC neurons have further been shown to encode context as well as task-related cues such as goal location (*Smith et al., 2012*; *Vedder et al., 2017*). Recent studies have focused on understanding multimodal integration (*Minderer et al., 2019*), accumulation of evidence (*Koay et al., 2019*), and how locomotion is differentially represented in RSC and visual cortex (*Clancy et al., 2019*). Finally, a subset of cells in RSC also encode head-direction in a way that is particularly sensitive to local environmental cues (*Jacob et al., 2017*). A common theme across these studies is the importance of visual inputs for RSC function. While the role of proximal, non-visual cues, such as whisker stimulation, has not been thoroughly evaluated, it is clear that visual cues alone are sufficient to guide behavior (*Etienne et al., 1996*). Together, these converging results strongly implicate RSC as an important neural substrate for landmark encoding.

We set out to identify how visual cues that inform an animal about a goal location are represented in RSC. We focus in particular on the dysgranular part of RSC (Brodmann Area 30) which is less well characterized compared to granular RSC (A29), but has been shown to express spatial receptive fields (*Mao et al., 2017*). We developed a task in which animals learn the spatial relationship between a salient visual cue and a hidden rewarded zone on a virtual linear track. The visual cue serves as a landmark indicative of the animal's distance to a reward. Studies investigating how spatial tuning is influenced by the environment generally use a single orienting cue (*Hafting et al., 2005*; *Muller and Kubie, 1987*; *Taube et al., 1990*), visual cues directly indicating the presence or absence of a reward (*Pakan et al., 2018*; *Poort et al., 2015*), or cue-rich environments where understanding the visual surrounding was not required to locate rewards (*Campbell et al., 2018*; *Fiser et al., 2016*; *Gauthier and Tank, 2018*; *Harvey et al., 2009*; *Saleem et al., 2018*). In contrast, our task requires mice to use allocentric inputs as reference points, and combine them with self-motion feedback to successfully execute trials. This task utilizes landmarks as an indicator of distance to a reward on a linear track, as opposed to as an orientation cue. While A30 does contain head-direction tuned neurons, they are unlikely to contribute to computations in this task. We found the majority of task-active neurons, as well as the population response, to be anchored by landmarks. Showing the same visual stimuli at a static flow speed while animals were not engaged in the task resulted in significantly degraded responses, suggesting that active navigation plays a crucial role in RSC function. Landmark responses were the result of supralinear integration of visual and motor components. To understand how visual information is translated into behaviorally-relevant representations in RSC, we recorded the activity of axons from the primary visual cortex (V1) during task execution. V1 sends strong projections to RSC (*Oh et al., 2014*) which, in turn, sends a top-down projection back to V1 (*Makino and Komiyama, 2015*), creating a poorly understood cortico-cortical feedback loop between a primary sensory and associative cortex. Understanding this circuit could provide key insights into how sensory and contextual information combine to guide behavior. We found strikingly similar receptive fields as those expressed by RSC neurons, suggesting that V1 inputs may be key in shaping their receptive fields. Importantly, their activity was less modulated by active navigation, illuminating a key difference between primary sensory and associative cortex.

## Results

### An RSC-dependent visual landmark navigation task

We developed a behavioral task that required mice to learn the spatial relationships between visual cues and hidden rewards along a virtual linear corridor (*Figure 1A and B*). Each trial began at a randomized distance (50–150 cm) from one of two salient visual cues with a vertical or diagonal stripe pattern respectively. Along the rest of the corridor, which appeared infinitely long, a gray-and-black dot uniform pattern provided optic flow feedback but no spatial information. An unmarked 20 cm wide reward zone was located at fixed distances from the visual landmarks (starting at 80 or 140 cm from the end of the landmark, respectively). At the end of each trial, the animal was 'teleported' into a 'black box' (black screens) for at least 3 s. A trial ended when an animal either triggered a reward by licking within the reward zone or received a 'default reward' when it passed the reward zone without licking. Default rewards were provided throughout the experiment but constituted only a small fraction of trials in trained animals (mean ± SEM: 14.24 ± 2.65% of trials, n = 12 sessions, 11 mice). The 3 s black box timeout was included in the task to give the animals a salient signal for the end and start of trials. It further ensured that GCaMP6f signals underlying different aspects of behavior (reward delivery and consumption versus the initiation of a new trial and changes in locomotion) could decay before the start of the next trial. Mice learned to use the visual cues to locate the reward zones along the corridor (*Figure 1C*, mean 32.3 ± 3.6 training sessions for n = 10 mice). In theory, animals could achieve a high fraction of successful trials by licking frequently but randomly, or in a uniform pattern. We tested if we could use licking as a behavioral assay for an animal's understanding of the spatial relationship between visual cue and reward location by calculating a spatial modulation index using a bootstrap shuffle test (*Figure 1—figure supplement 1E,F* and Materials and methods). This test randomly shifted licking locations relative to the track location for each trial and evaluated whether the animal would have still triggered a reward based on randomized licking (at least one lick inside the reward zone). The overall fraction of successful trials is then

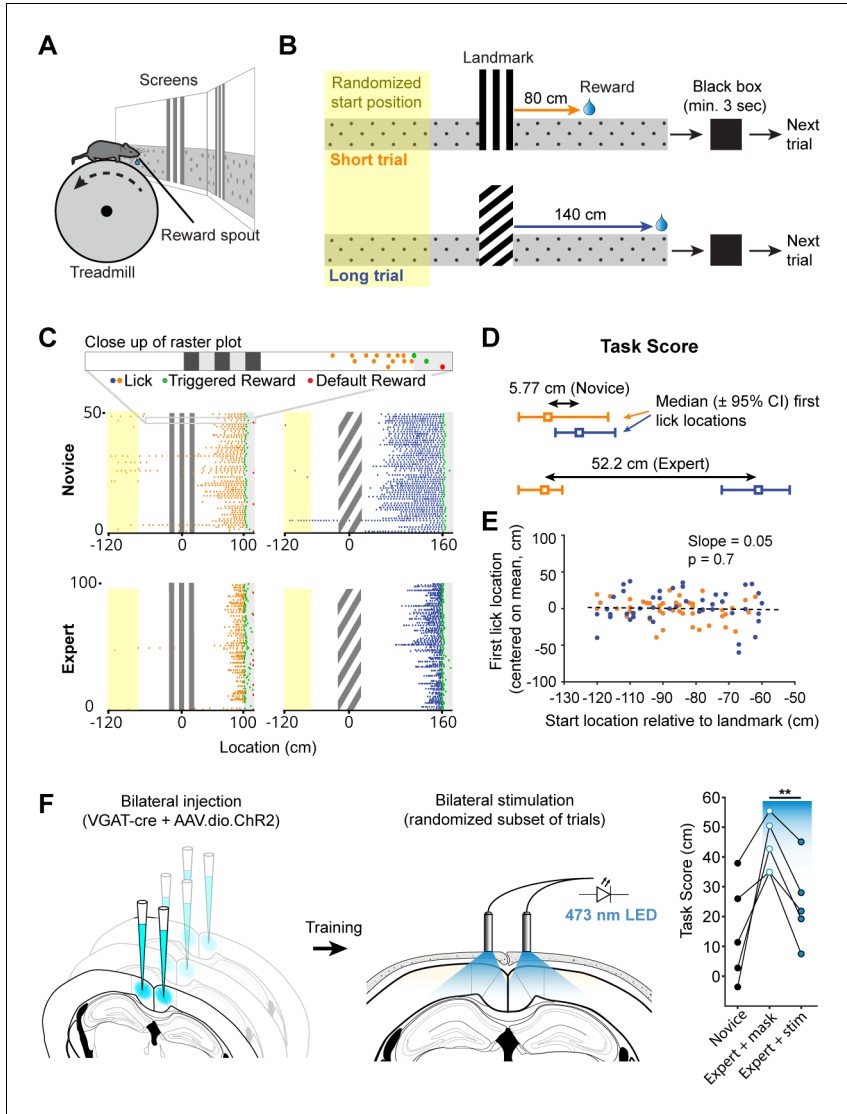

**Figure 1.** Landmark-dependent navigation task in virtual reality. (A) Schematic of experimental setup: mice are head-fixed atop a cylindrical treadmill with two computer screens covering most of the animal's field of view. A reward spout with attached lick-sensor delivers rewards. (B) Task design. Animals learned to locate hidden reward zones at a fixed distance from one of two salient visual cues acting as landmarks. The two landmarks were interleaved within a session, either randomly or in blocks of 5. After each trial animals were placed in a 'black box' (screens turn black) for at least 3 s. The randomized starting location ranged from 50 to 150 cm before the landmark. (C) Licking behavior of the same animal at novice and expert stage. Expert animals (bottom) lick close to the reward zones once they have learned the spatial relationship between the visual cue and reward location. (D) The Task Score was calculated as the difference in first lick location (averaged across trials) between short and long trials. (E) Relationship between trial start and first lick locations for one example session. Experimental design ensured that alternative strategies, such as using an internal odometer, could not be used to accurately find rewards. (F) RSC inactivation experiment. VGAT-Cre mice were injected with flexed Channelrhodopsin-2 (left). Stimulation light was delivered through skull-mounted ferrules on a random subset of trials (middle). During inactivation trials, task score was reduced significantly (right).

The online version of this article includes the following figure supplement(s) for figure 1:

**Figure supplement 1.** Running and licking behavior in naive animals and during optogenetic silencing.

calculated for the entire session using randomized licking locations. This process is repeated 1000 times. Finally, the z-score of the actual success rate relative to the shuffled distribution is calculated and checked whether it was significantly higher than the shuffled distribution. Licking behavior of expert animals was significantly spatially modulated (mean ± SEM: 13.48 ± 1.23, n = 12 session, 11 mice), making it an accurate behavioral assay.

We quantified an animal's ability to use landmarks for navigation by calculating the difference between the median location of first licks on short and long trials, expressed as a 'task score' (mean ± SEM of n = 12 recording sessions: 34.2 ± 5.47 cm). Location of first licks as opposed to mean lick location or lick frequency was used as it provided the most conservative measure of where animals anticipated rewards. This task structure inherently minimizes the ability of mice using alternative strategies such as time or an internal odometer to locate rewards. We tested whether animals used the start of the trial and a fixed distance, time, or number of steps before they started probing for rewards (*Figure 1E*). For each trial in one session, we plotted the location of the start vs. the location of the first lick and evaluated the linear regression coefficient showing no dependence of first lick on trial start location (mean ± SEM of slope: 0.13 ± 0.043W, n = 12 sessions).

To test if RSC was involved in task performance, we suppressed RSC activity by selectively activating inhibitory interneurons in expert animals during the task. We bilaterally injected a viral vector containing Cre-dependent channelrhodopsin-2 (ChR2) in multiple locations along the anterior-posterior axis of RSC (2–3 injections per hemisphere, *Figure 1—figure supplement 1H*) in VGAT-Cre mice. This restricted ChR2 expression to GABAergic neurons in RSC and allowed us to rapidly and reversibly inhibit local neural activity (*Lewis et al., 2015*; *Liu et al., 2014*). Ferrules were implanted on the surface of the skull over RSC to deliver light during behavior. Stimulation was delivered by a 470 nm fiber-coupled LED on a randomized 50% subset of trials. The stimulation light was turned on at the beginning of a given trial and lasted until the end of the trial or the maximum pulse duration of 10 s was reached. A masking light was shown throughout the session. Task scores on trials with stimulation was significantly lower compared to trials where only the masking light was shown within the same session (43.7 ± 7.4 cm vs. 24.3 ± 6.2 cm, n = 5 mice, paired t-test: p = 0.003, *Figure 1F*), indicating that RSC activity contributes to successful execution of this behavior. The fraction of successful trials, in contrast, was not significantly different in the mask only and stim condition (mask: 7.5 ± 2.0%, stim: 6.7 ± 1.9%, paired t-test: p=0.61), showing that the decrease in task score was not a result of a diminished ability to trigger rewards.

## Trial onset, landmark, and reward encoding neurons in RSC

We sought to understand which task features were represented by neurons in dysgranular RSC (A30). Mice injected with AAV expressing the genetically encoded calcium indicator GCaMP6f were trained until they reliably used landmarks to locate rewards. On average, we recorded 120.0 ± 17.56 RSC neurons per mouse (n = 7 mice). GCaMP signals of all active neurons (>0.5 transients/min, n = 966) were tested for significant peaks of their mean response above a shuffled distribution (z-score of mean trace >3, see Materials and methods) and for transients on at least 25% of trials. The calcium traces of individual neurons that met these criteria (n = 491) were grouped by trial type and aligned to each of three points: trial onset, landmark, and reward (*Figure 2A*, right; *Figure 2B*). The peak activity of the mean GCaMP trace of each neuron was then compared across alignment points and classified based on which task feature resulted in the largest mean response (*Figure 2B*). This analysis was carried out for short trials and long trials independently. The majority of RSC neurons found to be task engaged were aligned to the visual landmark (*Figure 2C*, n = 55 short trial onset and 62 long trial onset, 206 and 235 landmark, 101 and 118 reward neurons, respectively; seven mice; mean ± SEM fractions of aligned neurons: trial onset: 5.6 ± 0.3%, landmark: 22.3 ± 1.6%, reward: 11.6 ± 0.7%; one way ANOVA$_{short}$: p = 0.0023; ANOVA$_{long}$: p<0.001, Tukey's HSD post-hoc pairwise comparison with Bonferroni correction). A smaller but sizeable fraction of RSC neurons were aligned to the reward point, suggesting that RSC also encodes behavioral goals as well, and the onset of a trial, regardless of where an animal was placed on the track relative to the visual landmark. Consistent with previous findings (*Alexander and Nitz, 2015*), these data indicate that egocentric (trial onset) as well as allocentric (landmark and reward) variables are encoded in RSC during landmark-based navigation. The vast majority of neurons showed a single peak of activity in our task. Previous studies have found neurons with multiple peaks or sustained firing in RSC neurons (*Alexander and Nitz, 2015*; *Alexander and Nitz, 2017*). Our task, however, did not contain

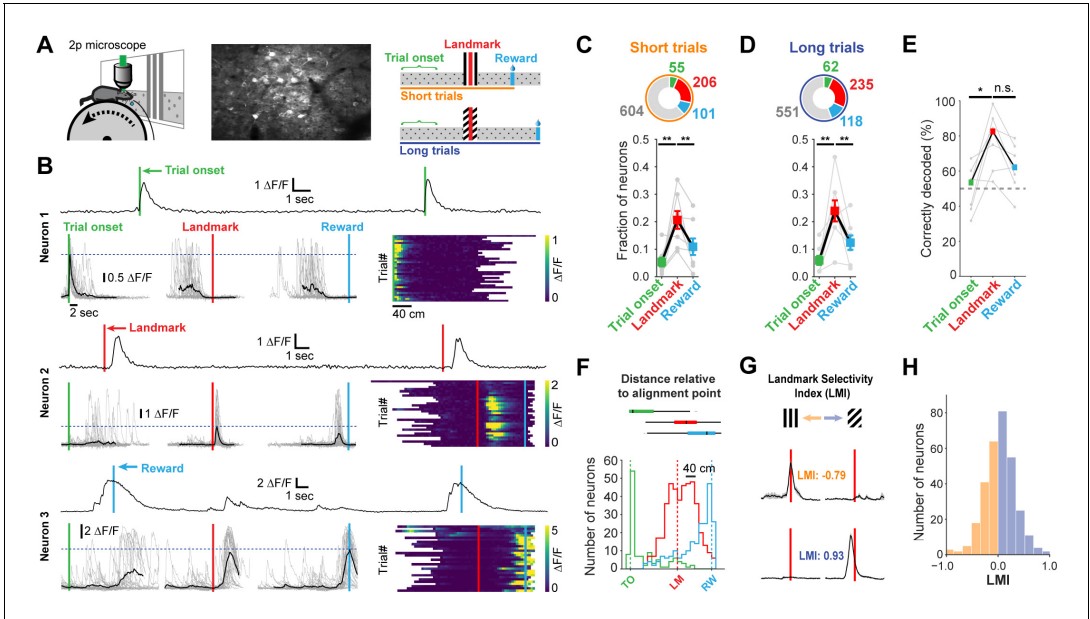

**Figure 2.** Neuronal responses in RSC during landmark-dependent navigation. (**A**) Left: schematic of recording setup. Middle: example field of view. Right: Alignment points and trial types. The activity of each neuron was then aligned to each of three points: trial onset (green), landmark (red), and reward (cyan). Responses were independently analysed for short and long trials. (**B**) Each neuron's best alignment point was assessed by quantifying the peak of its mean trace and comparing it to the other alignment points. Rows show trial onset (top), landmark (middle), and reward aligned (bottom) example neurons. (**C, D**) Alignment of task-active neurons. The majority of task-engaged neurons were aligned to the landmark on short (**C**) and long (**D**) trials (n = 7 mice). (**E**) We applied a template matching decoder (*Montijn et al., 2014*) to decode the trial type based on the neural responses recorded from each animal. Trial onset neurons provided chance level decoding. However, landmark neurons provided significantly higher decoding accuracy which remained elevated for reward neurons (**F**) Mean distance of transient peaks of individual neurons relative to alignment point. (**G**) Two landmark-selective neurons. Landmark selectivity was calculated as the normalized difference between peak mean responses. (**H**) The landmark selectivity index (LMI) of all landmark neurons shows a unimodal distribution.

The online version of this article includes the following figure supplement(s) for figure 2:

**Figure supplement 1.** Differences in landmark encoding in layer 2/3 and layer 5.
**Figure supplement 2.** Simultaneous imaging of layer 2/3 and 5.

repeating sections found in a W-shaped or plus-shaped maze which may explain the discrepancy in our findings. The peak response of a given neuron did not necessarily need to happen directly at a given alignment point, but could also occur at some distance from it. A landmark-aligned neuron, for example, did not have to exhibit its peak response at the landmark. Instead, it could be active close to the reward zone but its calcium trace was still best aligned to the landmark (*Figure 2F*).

Surprisingly, we found no significant difference in transient amplitude or robustness (probability of a transient) between trials where mice operantly triggered rewards themselves compared to 'unsuccessful' trials, in which they received the default reward across all neuron types (mean amplitude on successful trials: 1.79 ± 0.09 ΔF/F, unsuccessful trials: 1.69 ± 0.09 ΔF/F, n = 241 neurons, paired t-test: p=0.9. Mean robustness on successful trials: 0.61 ± 0.02, unsuccessful trials: 0.61 ± 0.09, n = 210 neurons, paired t-test: p=0.07. This analysis only included neurons in sessions with successful and unsuccessful trials on short or long track). However, we found neurons that were differentially active within a session when we split the responses into the most and least accurate 25% of trials. A subset (21.6%) changed their activity by >0.5 ΔF/F. These changes were bidirectional: some neurons increased their activity while others decreased their activity based on how well an animal predicted the location of a reward, measured as the distance of the first lick in a given trial to the start of the reward zone (*Figure 4—figure supplement 1C,D*). We employed a template matching decoder (*Montijn et al., 2014*) to analyze how well trial type could be decoded from neural activity alone (*Figure 2E*). While trial onset or reward-aligned neurons provided only chance level decoding or slightly better (trial onset: median 53.44% correct; reward: 62.1%), trial type decoding

by landmark neurons was significantly higher (82.56%, Kruskal-Wallis test p=0.015; post-hoc Mann-Whitney U pairwise testing with Bonferroni correction for multiple comparisons).

We examined how individual task features (landmark, trial onset, and rewards) are differentially represented in RSC layers 2/3 (L2/3) and layer 5 (L5). Cortical layers were identified by their depth under the dura (L2/3: 130.0 ± 4.0 µm, L5: 327.0 ± 15.2), and confirmed post-hoc with histological sections in a subset of animals (*Figure 2—figure supplement 1A*, *Figure 2—figure supplement 2A*). We found that superficial as well as deep layers contained trial onset, landmark, and reward neurons. However, L5 contained substantially fewer landmark neurons (*Figure 2—figure supplement 1B,C*; One-way ANOVA, p<10–8, Tukey HSD post-hoc test with Bonferroni correction).

Finally, we asked whether the subpopulation of landmark encoding neurons showed a preference for visual cue identity. We calculated a landmark modulation index as the difference between peak activity divided by the sum of their activity [$LMI = (LM_{short} - LM_{long})/(LM_{short} + LM_{long})$]. Peak activity for each trial type was calculated separately. Only a small number of neurons were found to be tuned to landmark identity (*Figure 2H*), with most neurons showing no specific preference. Similarly, trial onset neurons and reward neurons did not show trial type selectivity (*Figure 3—figure supplement 1E*). These results indicate that neurons in RSC encode a mix of task variables with a strong preference for visual cues informing the animal about goal locations.

## Landmarks anchor the representation of space in RSC

In this task, trial onset and visual landmarks provide egocentric and allocentric context, respectively. We tested which reference point anchored the neural representation of the animal's location using population vector cross-correlation analyses for all task active neurons (n = 491, *Gothard et al., 1996a*; *Alexander and Nitz, 2015*; *Alexander and Nitz, 2017*; *Mao et al., 2017*). Two activity vectors were constructed for each neuron by randomly taking data from one half of the trials for the first vector and the other half for the second vector. Data from the first vector was used to determine the location of a neuron's maximum response while the second vector was used for correlation analysis. This process prevents introduction of artifactual spatial structure into the population code. Activity was binned into 5 cm wide bins and the mean across all included trials was calculated and normalized to 1. We found largely even tiling of space from the trial onset until reward (*Figure 3A and D*). To test the spatial specificity of the population code, we calculated a population vector cross-correlation matrix using the Pearson cross-correlation coefficient (*Figure 3B and E*) for each spatial bin. To ensure that randomly splitting data into halves didn't lead to spurious results, we calculated the mean of 10 cross-correlation maps, each randomly drawing a different subset of trials. Slices of the cross-correlation matrix (*Figure 3C and F*, taken at the dashed lines indicated in 3B and 3E), reveal that the spatial code is sharpest at the landmark. The cross-correlation at the animal's true location (i.e. along the diagonal from top left to bottom right) significantly increases as the animal approaches the landmark and remains elevated until it reaches the reward (*Figure 3G*). We tested how well we could reconstruct the animal's location from the neural code by calculating how far the pixel with the highest cross-correlation was from the actual location for each row in the cross-correlation matrix. We observed a significantly lower location reconstruction error when neural activity was aligned to landmarks, rather than trial onset (*Figure 3H*, mean ± SEM: 3.7 ± 0.61 vs. 5.27 ± 0.56 short trials; 3.23 ± 0.48 vs. 5.26 ± 0.42, unpaired, 2-tailed t-test: p<0.016 (short), p<0.001 (long)). Finally, we found no significant difference in reconstruction error between short track and long track trials, as most neurons are active on both trials (*Figure 3—figure supplement 1*). These results provide evidence for a spatial code in RSC that is strongly modulated by environmental cues to inform the animal about the location of its goal.

## Active task execution sharpens spatial tuning and increases robustness of responses in RSC

To determine if goal-directed navigation, as opposed to visual input alone, was required to explain RSC activity, we recorded neurons while animals were shown the same virtual corridor without actively executing the task. During this decoupled stimulus presentation paradigm (DC), the virtual corridor moved past the animals at two speeds: 10 cm/sec and 30 cm/sec (*Figure 4A*, see *Figure 1—figure supplement 1B,D* for speed profiles during virtual navigation). Both trial types were interleaved in the pattern as during virtual navigation, but no rewards were dispensed when animals

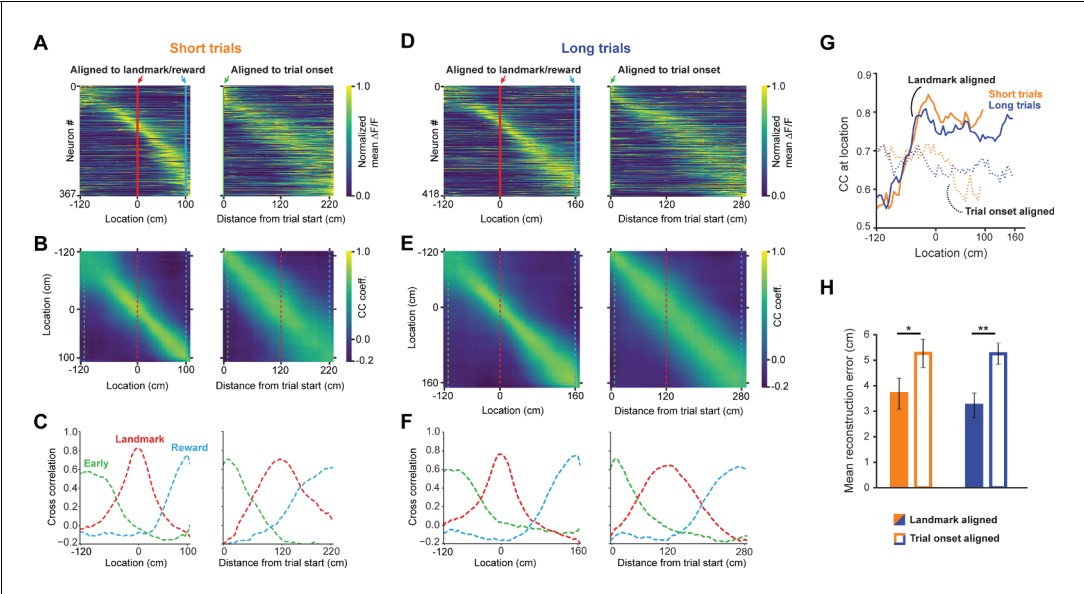

**Figure 3.** A landmark-anchored code for space in RSC. (**A**) Activity of all task engaged neurons ordered by location of peak activity on short trials. Left columns: neurons aligned to the landmark/reward; right columns: same neurons aligned to trial onset point. (**B**) Population vector cross-correlation matrices of data shown in (**A**). (**C**) Slices of the cross-correlation matrices early on the track (green dashed line), at the landmark (red dashed line), and at the reward point (blue dashed line), show sharpening of the spatial code at the landmark. (**D–F**) Same as (**A–C**) but for long trials. (**G**) Population vector cross-correlation values at the animal's actual location. Solid lines: activity aligned to landmark/reward; dashed lines: activity aligned to trial onset. (**H**) Reconstruction error, calculated as the mean distance between the maximum correlation value in the cross-correlation matrices and the animal's actual location, is significantly lower when neural activity is aligned to landmarks (solid bars) compared to trial onset aligned (open bars; Mann-Whitney U: short trials: p<0.05, long trials: p<0.001).

The online version of this article includes the following figure supplement(s) for figure 3:

**Figure supplement 1.** Short track and long track neurons activity on opposite tracks.

reached reward locations. We found a significant decrease in neuronal responses in this condition (**Figure 4B and C**, mean ± SEM VR: 0.17 ± 0.005, mean decoupled: 0.059 ± 0.004, paired, two-tailed t-test: p<0.001). This was true for neurons of all three categories: trial onset, landmark, and reward (**Figure 4D**, median values VR: trial onset = 0.1, landmark = 0.19, reward: 0.13; decoupled: trial onset = 0.03, landmark = 0.04, reward = 0.02). This result suggests that activity in RSC is strongly dependent on active task engagement. Congruent with this, population activity showed significantly less spatial specificity during decoupled stimulus presentation (**Figure 4E–H**, mean reconstruction error ± SEM: 3.7 ± 0.61 vs. 7.88 ± 0.78 short trials; 3.23 ± 0.48 vs. 11.16 ± 1.18, unpaired, Mann-Whitney U: p<0.001 (short), p<0.001 (long)). This indicates that encoding of behaviorally-relevant variables in RSC is modulated by ongoing behavior, rather than being driven solely by sensory inputs. We note that not providing a reward and decoupling the stimulus presentation from animal locomotion constitute simultaneous changes that may both influence neural activity. However, if reward anticipation was a key driver in the change in neural activity we would expect neurons anchored by trial onset or landmark to be less affected than reward driven neurons. We find that all neuron types are similarly affected (**Figure 4D**), suggesting that reward anticipation is not the major cause for the change in activity we observed. A second potential factor modulating neuronal responses is whether the animal is attending to the cue or not. We have addressed this issue by analyzing responses during quiet wakefulness and locomotion in the next section (**Figure 5**). Finally, attending to the stimuli and/or task may significantly impact the respective neural representation. We therefore recorded pupil dilation in three well trained animals. However, we found no significant difference in pupil dilation during active navigation and decoupled stimulus presentation (**Figure 4— figure supplement 1F–H**), suggesting that attention is an unlikely explanation for our results. We sought to gain insight into potential mechanisms underlying the changes in neuronal activity during decoupled stimulus presentation by comparing GCaMP6f signals observed in both conditions.

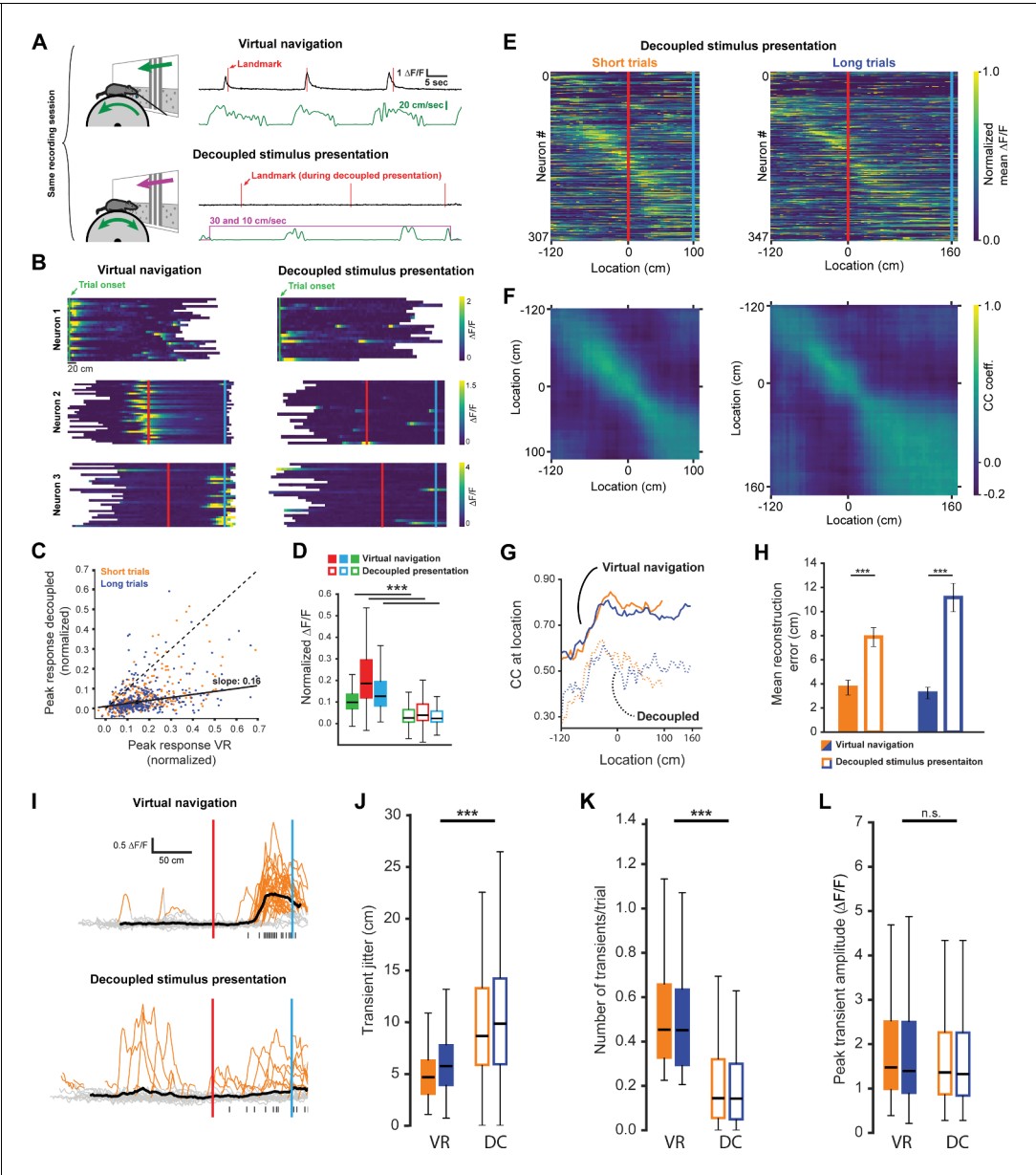

**Figure 4.** Neuronal activity during decoupled stimulus presentation. (**A**) Recording session structure: after recording from neurons during virtual navigation, the same stimuli were presented in an 'open loop' configuration where the flow speed of the virtual environment was decoupled from the animal's movement on the treadmill. (**B**) Trial onset, landmark, and reward example neurons under these two conditions. (**C, D**) Response amplitudes of all task engaged RSC neurons during decoupled stimulus presentation (Kruskal-Wallis: p<0.0001, Wilcoxon signed-rank pairwise comparison with Bonferroni correction indicated in (**D**)). (**E, F**) Population activity and population vector cross-correlation during decoupled stimulus presentation for short (left) and long (right) trials. (**G**) Local cross-correlation at animal's location is smaller during decoupled stimulus presentation. (**H**) Mean location reconstruction error. Reconstructing animal location from population vectors is significantly less accurate when the animal is not actively navigating (unpaired t-test, short trials and long trials: p<0.0001). (**I**) Traces of example neuron activity overlaid during virtual navigation (top) and decoupled stimulus presentation (bottom) with transients highlighted in orange. Ticks along bottom indicate peaks of transients around the neuron's peak response. (**J**) Spread of transient peak location around peak mean response measured as the standard error of the mean of (standard error of mean of transient peak location – mean peak location). Solid bars: virtual navigation (VR), open bars: decoupled stimulus presentation (DC). (**K**) Average number of transients/trial during virtual navigation and decoupled stimulus presentation. (**L**) Average amplitude of transients in VR and DC conditions. Boxplots show median, 1st - 3rd quartile, and 1.5 interquartile range.

The online version of this article includes the following figure supplement(s) for figure 4:

**Figure supplement 1.** Neural correlates of running speed and reward location prediction.

**Figure supplement 2.** Stable long-term imaging.

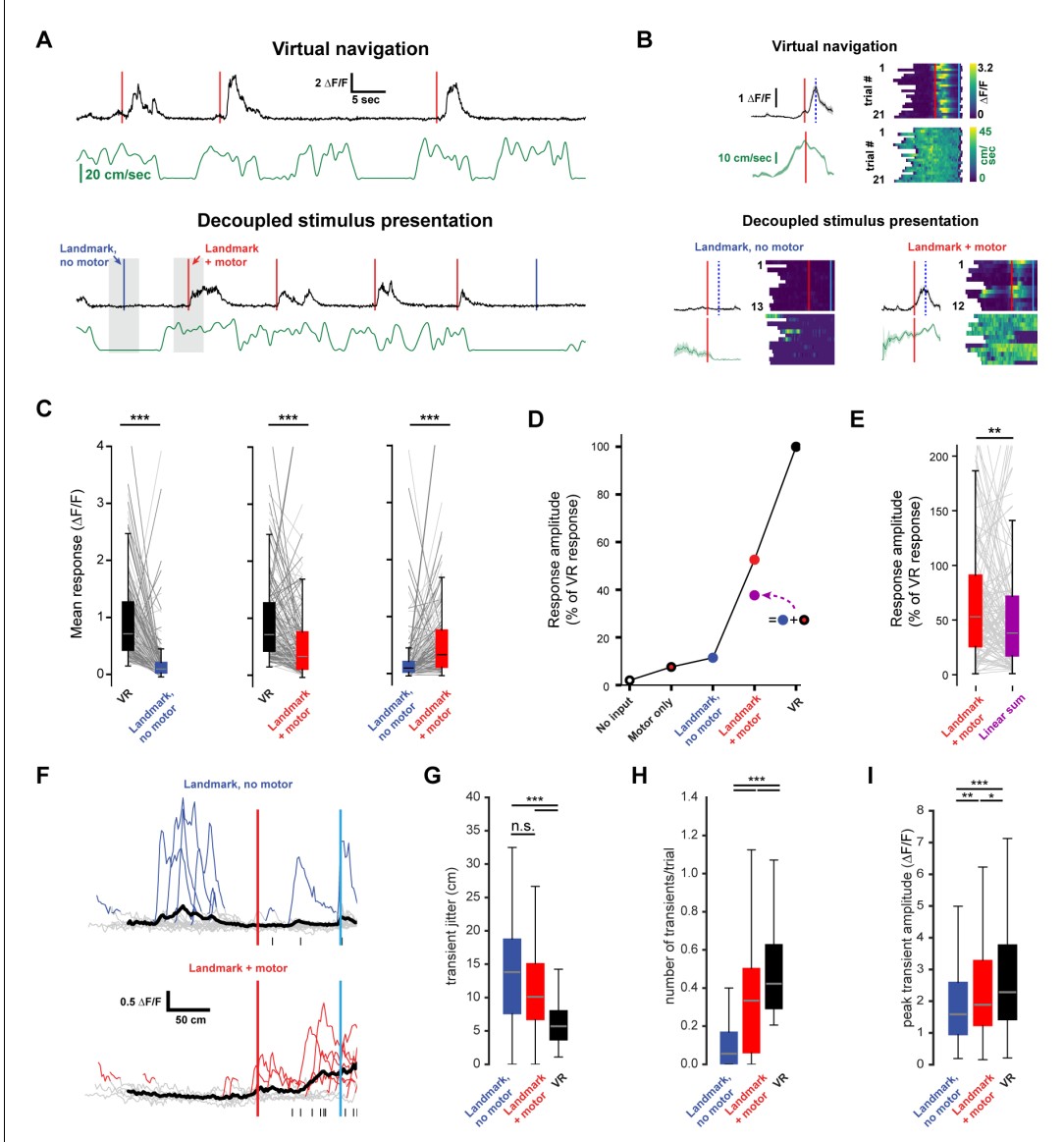

**Figure 5.** Non-linear integration of visual and motor inputs in RSC landmark neurons. (**A**) Example neuron during virtual navigation (top) and decoupled stimulus presentation as the animal is running or resting (bottom). (**B**) Same neuron as in (**A**) with all instances where the animal was running or resting, averaged (left) and raster plots of the whole session (right). Peak mean activity indicated by dashed blue line. (**C**) Activity of population during virtual navigation and decoupled stimulus presentation (**D**) Neuronal responses normalized to peak activity in VR under different conditions. 'No input' and 'motor only' responses were measured while animals are in the black box between trials (median and spread of data shown in (**C**)). (**E**) The sum of 'landmark, no motor' + 'motor' is smaller than 'landmark + motor' responses suggesting nonlinear combination of visual and motor inputs. (Wilcoxon signed-rank test, p<0.01). (**F**) Traces of example neuron when the animal is passively watching the scene (top) or locomoting (bottom). Black ticks along bottom indicate transients around that neuron's peak mean activity during virtual navigation (see **Figure 4I**). (**G**) Spread of transient location around peak mean activity in VR (standard error of mean of transient peak location – mean peak location). (**H**) Average number of transients/trial and (**I**) average amplitude in both conditions. Kruskal-Wallis test p<0.0001; Mann-Whitney-U pairwise comparisons with Bonferroni correction results indicated, * = <0.05, ** = <0.01, *** = <0.001. Boxplots show median, 1st - 3rd quartile, and 1.5 interquartile range.

Individual events were detected when ΔF/F exceeded six standard deviations of a neuron's baseline activity for at least two spatial bins (bin size: 2 cm) and lay within ±60 cm of the peak mean response (**Figure 4I**). We found that the standard error of the distance of individual transients to the peak of the mean trace (**Figure 4J**, median jitter (cm): $short_{VR}$ = 4.67, $short_{DC}$ = 8.66; $long_{VR}$ = 5.75, $long_{DC}$ = 9.85) was lower during virtual navigation compared to decoupled stimulus presentation. In other words, transients were more tightly clustered around that neuron's peak response when the

animal was actively engaged in the task. Furthermore, we saw a significant reduction in the number of transients per trial during decoupled stimulus presentation (*Figure 4K*, median values (transients/trial): short$_{VR}$ = 0.45, short$_{DC}$ = 0.14; long$_{VR}$ = 0.45, long$_{DC}$ = 0.14), but only very little change in the amplitude of individual transients (*Figure 4L*, median values ($\Delta$F/F): short$_{VR}$ = 1.48, short$_{DC}$ = 1.36; long$_{VR}$ = 1.4, long$_{DC}$ = 1.33, Kruskal-Wallis test p<0.001; Mann-Whitney-U pairwise comparisons with Bonferroni correction results indicated, ***=p < 0.001). These results show that the changes during decoupled stimulus presentation is due to poorer spatial anchoring of activity and fewer instances of a given neuron to exhibit a transient.

## Nonlinear integration of visual and motor inputs in RSC

Nonlinear integration of synaptic inputs dramatically enhances the computational power of individual neurons and neural networks (*Miller and Cohen, 2001*; *London and Häusser, 2005*; *Mante et al., 2013*; *Rigotti et al., 2013*; *Jadi et al., 2014*; *Ranganathan et al., 2018*).

For a single neuron, the integration of multiple input streams may engage mechanisms of supra-linear integration to produce complex, conjunctive responses (*Bittner et al., 2015*; *Larkum et al., 1999*; *Takahashi et al., 2016*; *Xu et al., 2012*). In contrast, neural networks may express conjunctive representations through high dimensional codes (*Murray et al., 2017*; *Rigotti et al., 2013*; *Stringer et al., 2019*). We therefore evaluated the evidence for nonlinear integration in landmark-anchored RSC neurons. During decoupled stimulus presentation, mice were free to spontaneously locomote on the treadmill or watch passively. Trials within a session were separated based on whether the animal locomoted as the virtual environment passed a neuron's receptive field (*Figure 5A and B*, '+ motor' trials, running speed >3 cm/sec in a ± 50 cm window). 'No input' and 'motor only' conditions were measured while the animal was in the black box in between trials using similar criteria as before but with a 1.5 s time window (instead of a spatial window) for locomotion. Thus, any brief spontaneous movements that may have occurred within that window, but were below threshold, were labeled passive viewing. Despite the possibility that such small movements impact neural responses, we find a striking contrast between locomoting and passively viewing animals. When landmark presentation occurred during locomotion, activity was significantly increased (*Figure 5C*, n = 127 neurons, five mice, mean ± SEM number of trials/neuron: 13.0 ± 0.41 resting, 9.3 ± 0.3 running; peak mean $\Delta$F/F ± SEM: VR: 0.93 ± 0.06, landmark + motor: 0.57 ± 0.06, landmark, no motor: 0.22 ± 0.04, Kruskal-Wallis test: p<0.001, Mann-Whitney U post-hoc comparison with Bonferroni correction: p<0.001 for all shown comparisons). However, we found that visual inputs alone or visual inputs plus locomotion did not elicit the same response as virtual navigation (*Figure 5C,D*, Kruskal-Wallis test: p<0.001; post-hoc Mann-Whitney-U and Bonferroni correction: p<0.001 for all shown comparisons). We then evaluated the responses while animals were locomoting or stationary while in the 'black box' between trials to obtain estimates of population activity in 'no input' (neither visual nor motor inputs) and 'motor only' conditions (*Figure 5D*, $\Delta$F/F ± SEM: black box + motor = 0.16 ± 0.03, black box, no motor = −0.04 ± 0.03). Finally we asked whether the linear sum of 'motor only' and 'landmark, no motor' added up to 'landmark + motor' (*Figure 5D and E*). We found that the linear sum approached, but remained lower than the mean amplitude recorded during 'landmark + motor'. Analysis of GCaMP transient patterns during 'no motor' and '+ motor' conditions revealed that neurons show significantly more transients while locomoting (*Figure 5H*, median values (transients/trial): no motor = 0.06, + motor = 0.33; VR = 0.42), however, transient amplitude and jitter were broadly similar (*Figure 5G and I*, median jitter (cm): no motor = 13.82, + motor = 10.7; VR = 5.69; median amplitude ($\Delta$F/F): no motor = 1.59, + motor = 1.89; VR = 2.28). Our results suggest that motor input drives RSC neurons, however it does not aid in anchoring their activity or modulating the number of spikes produced once it has reached firing threshold. We note that GCaMP6f may not provide linear translation from underlying spikes to fluorescence signal. However, our analysis focuses on relative differences within the same neuron under different conditions and thus nonlinearities of the calcium indicator are unlikely explain these results. Together these results provide evidence for substantial nonlinear integration of visual and motor inputs in RSC neurons during goal-directed virtual navigation as well as decreased, but still significant, nonlinear processing during decoupled stimulus presentation.

A possible explanation for this result is a correlation of between running speed and transient amplitude. To test this, we analyzed the modulation of transient amplitude by running speed (*Figure 4—figure supplement 1A,B*). We found only a small number of neurons showing modulation of

transient amplitude by running speed across the population of task-active neurons (*Figure 4—figure supplements 1B*, 0.5% on short trials n = 29 neurons and 15.6% on long trials n = 51 neurons). Of these neurons, 22 were positively and 58 negatively correlated. As we find changes in the vast majority of landmark-anchored neurons (*Figure 5C*), it is unlikely that running speed modulation explains these results.

## Changes in spatial encoding after task acquisition

During decoupled stimulus presentation, locomotion no longer influences the virtual environment or reward availability, which may change animals' internal state. We therefore performed experiments imaging the same neurons before and after learning in a separate group of 3 mice (mean ± SEM task score on expert session: 47.21 ± 2.6); animals were required to perform the same behavior, with the same stimulus and reward contingencies, in both conditions. We found some neurons that previously had not expressed discernible spatially tuned activity, establishing spatial receptive fields, while showed amplified responses (*Figure 6A and B*). When we identified task active neurons in expert animals and calculated the population vector cross correlation during naïve and expert sessions, we found that the encoding of space in the naïve animal was much degraded (*Figure 6C–E*). However, this result could be explained by different subsets of neurons being active in naïve and expert animals, as opposed to a robust spatial code emerging during learning. To test this, we identified task active neurons in naïve and expert sessions independently and calculated the population vector cross correlation. Neurons in the naïve animal showed significantly worse representation of the animal's location (*Figure 6E*). The observed neural activity in expert animals is thus the result of a code that develops in RSC as animals learn to associate the visual cue with a reward location, turning it into a spatial landmark.

## V1 inputs to RSC represent task features but are less modulated by active navigation

Finally, we sought to dissect how dysgranular RSC produces landmark representations by identifying what information it receives from primary visual cortex (V1), a major input source to RSC (*Vogt and Miller, 1983*). To this end, we injected GCaMP6f into V1 in a separate group of trained animals and recorded the responses of axonal boutons in RSC (*Figure 7A and B*). Use of a passive pulse splitter (*Ji et al., 2008*) in the laser path allowed us to image axons continuously during self-paced behavioral sessions with no photobleaching or toxicity. To prevent overrepresentation of axons with multiple boutons in a given FOV, highly cross-correlated boutons were collapsed and represented as a single data point (see Materials and methods and *Figure 7—figure supplement 1*). In a separately injected and trained group of 4 animals we found a total of 77 unique, task-active axons. Unexpectedly, we found receptive fields that were strikingly similar to those we observed in RSC neurons (*Figure 7C and D*). Boutons also tiled space along the virtual linear track in a parallel manner to RSC neurons (*Figure 7E and F*). Furthermore, we found a similar preference of V1 boutons to be anchored by landmarks (*Figure 7G*). However, when we quantified how active task engagement modulates activity in RSC neurons versus V1 boutons, we found that the former were significantly more modulated compared to the latter (*Figure 7H and I*, Mann-Whitney U: p<0.001). These results are consistent with a number of recent studies that describe the encoding of non-visual stimuli in V1 (*Ji and Wilson, 2007*; *Niell and Stryker, 2010*; *Pakan et al., 2018*; *Poort et al., 2015*; *Saleem et al., 2018*). The specificity of these responses suggests that at least a subpopulation of RSC-projecting neurons in V1 is tuned to behaviorally-relevant visual cues to a previously unknown extent. Despite their specificity, however, they represent visual inputs more faithfully and are less modulated by context than RSC neurons themselves, pointing to the local computations performed in RSC.

## Discussion

In this study, we introduce a novel behavioral paradigm in which mice learned the spatial relationships between salient environmental cues and goal locations (*Figure 1*). The task required animals to discriminate visual cues, use them to localize themselves in space, and navigate to a rewarded zone based on self-motion feedback. Using this paradigm, we found that landmarks anchored the majority of task-active neurons in dysgranular RSC (*Figure 2C and D*) and significantly sharpened the

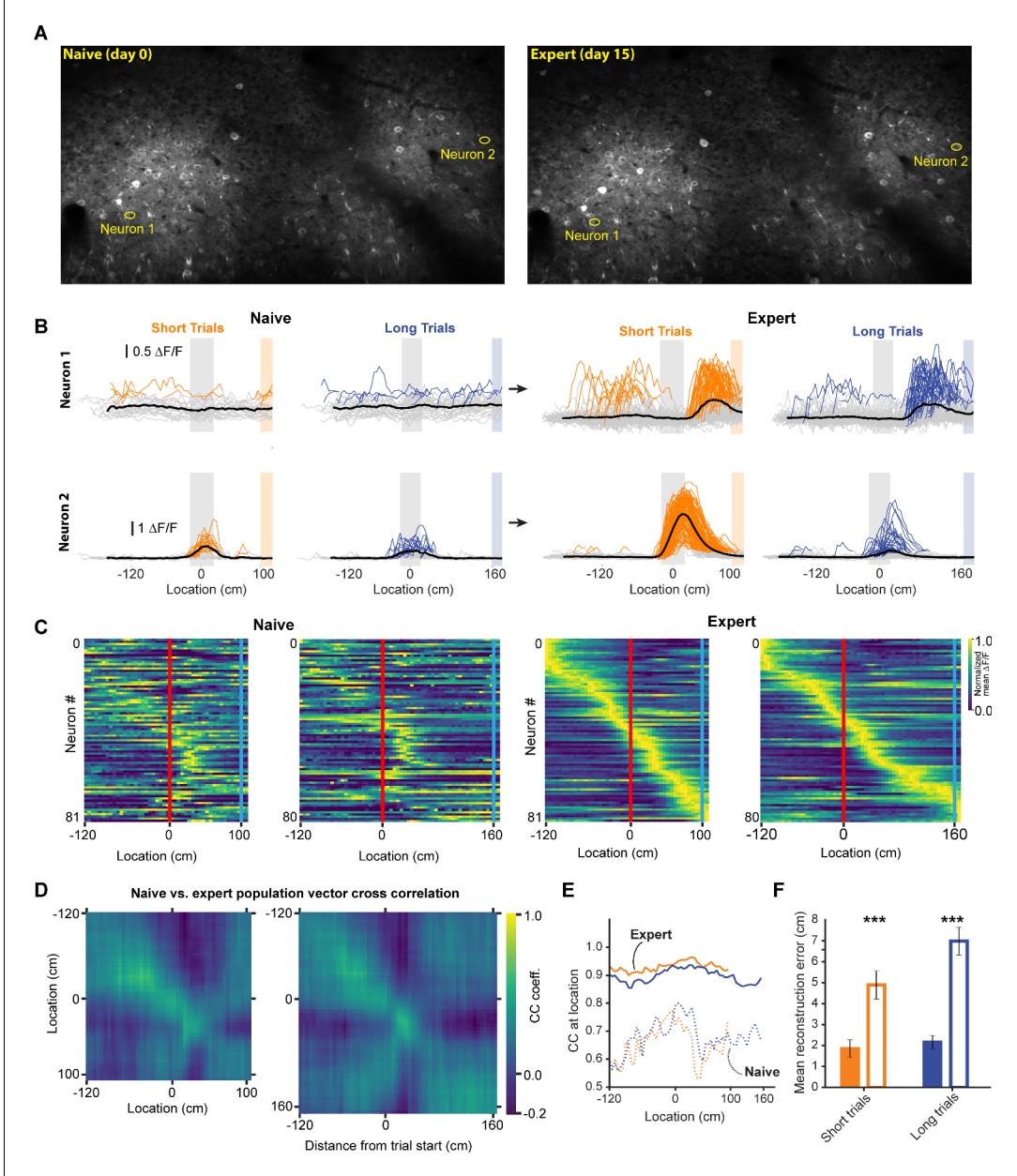

**Figure 6.** Neural activity in naive and expert animals. (**A**) Example field of views in the naïve (left) and expert (right) mouse with neurons that are shown in (**B**). Naïve mice were exposed to the virtual track for the first time after being previously habituated to being head restrained and running on a treadmill as well as receiving rewards at pseudo-random intervals. (**B**) Two example neurons that modified their activity as a function of training. Neuron 1 (top) showed no discernable receptive field in the naïve animal. However, in the expert animal, it showed a clear receptive field after the landmark. Neuron 2 (bottom) in contrast, showed some landmark-anchored activity that was strongly amplified in the expert condition. (**C**) Activity of task-active neurons in expert animals shown in both naïve and expert sessions (n = 81 short track, 80 long track). (**D**) Population vector cross-correlation matrix of activity in naïve and expert sessions. (**E**) Cross correlation value of population vectors at the actual location of the animal in naïve and expert sessions calculated from task active neurons in the respective sessions. (**F**) Reconstruction error in naïve and expert conditions (mean ± SEM reconstruction error short/long: 2.07 ± 0.37/2.67 ± 0.46 (expert), 5.1 ± 0.63/5.7 ± 0.62 (naive); two tailed t-test, short and long: p<0.001).

representation of the animal's current location in the population code (*Figure 3*). This spatial representation largely changed 2during learning (*Figure 6*). Landmark responses were not the result of simple visual and/or motor drive: showing the same visual stimuli while the animals were not engaged in the task elicited significantly attenuated responses (*Figure 4*). Further dissection of neuronal activity provided evidence for supralinear integration of visual and motor information in RSC.

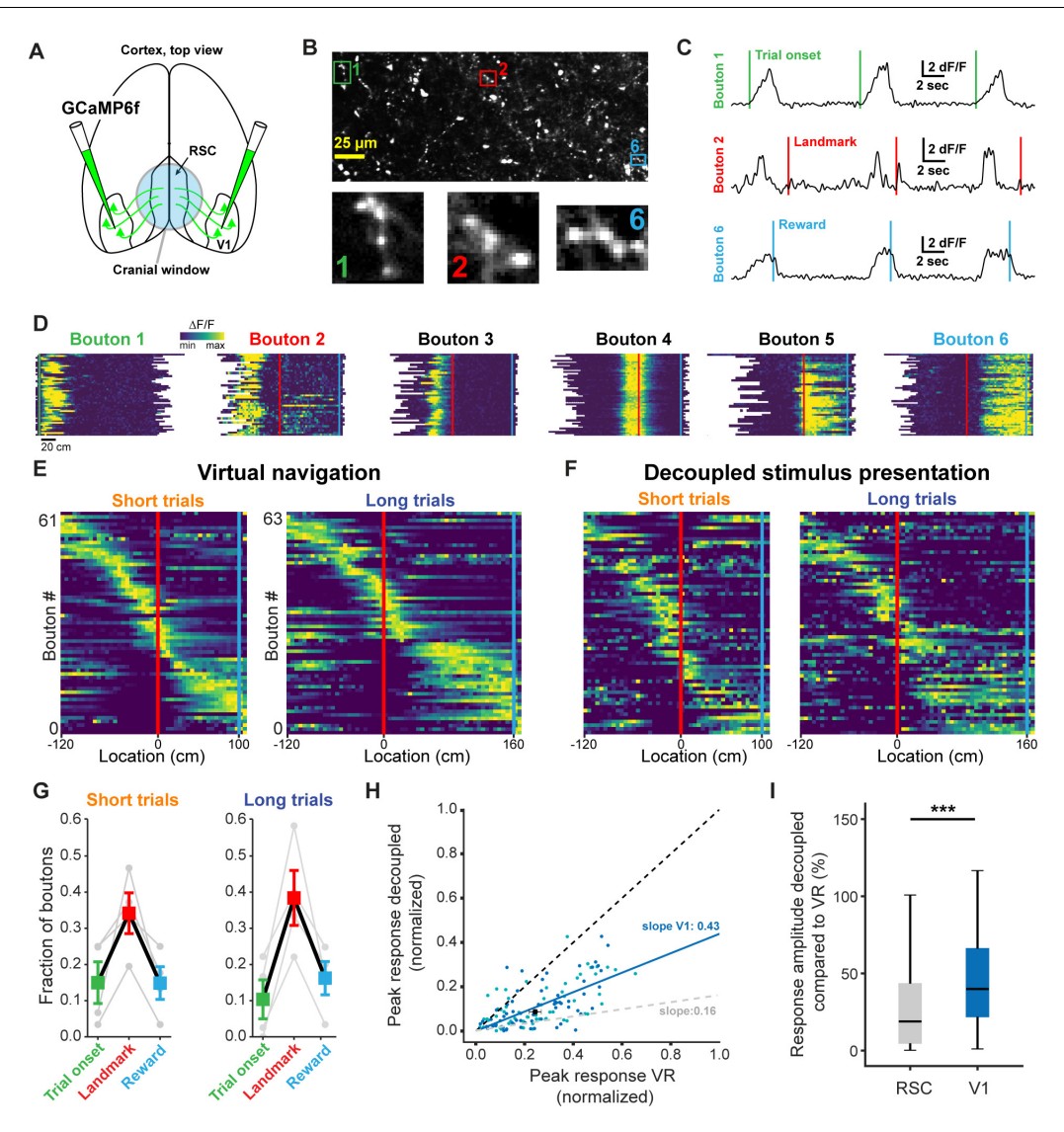

**Figure 7.** V1 axonal bouton responses in RSC. (**A**) Overview of injection and recording site. (**B**) Example FOV and three example boutons shown in (**C**) and (**D**). Where possible, ROIs were drawn around clusters of boutons belonging to the same axon. (**C**) Trial onset, landmark, and reward aligned boutons from same animal. (**D**) Six example boutons showing tuning to pre- and post-landmark portions of the track. (**E**) Population of V1 boutons in RSC ordered by location of their response peak (n = 61 boutons short track/63 boutons long track, four mice). (**F**) Same boutons as in (**E**) during decoupled stimulus presentation. (**G**) Alignment of boutons to task features. (**H**) Response amplitude during virtual navigation of decoupled stimulus presentation with fitted regression line. In gray: fitted regression line for RSC neurons. (**I**) Comparison of response amplitude differences between VR and decoupled stimulus presentation in RSC neurons and V1 boutons (mean$_{RSC}$ = 0.42 ± 0.03, mean$_{V1}$ = 0.53 ± 0.04, Mann-Whitney U test: p<0.001). The online version of this article includes the following figure supplement(s) for figure 7:

**Figure supplement 1.** Identification process for unique axonal inputs.

Coinciding visual input and motor feedback during decoupled stimulus presentation did not elicit the same response amplitudes as observed during active navigation (*Figure 5*). Interestingly, receptive fields expressed by V1 axonal boutons in animals executing the same behavior were strikingly similar to those recorded from RSC neurons (*Figure 7*). However, they were less modulated by active task engagement (*Figure 7H and I*), indicating a hierarchy of sequential processing.

A major challenge in understanding how computations in RSC contribute to behavior is the diversity and complexity of functions attributed to this area (*Maguire, 2001*; *Vann et al., 2009*). Studies in humans suggest that RSC is key for utilizing environmental cues during navigation (*Epstein, 2008*;

*Ino et al., 2007*; *Maguire, 2001*), while experiments in rodents found deficits in path integration (navigation based on self-motion cues) when RSC was lesioned or inactivated (*Cooper et al., 2001*; *Cooper and Mizumori, 1999*; *Elduayen and Save, 2014*). We describe a task that combines both of these navigational strategies: animals are required to use visual landmarks for self-localization, followed by path integration to successfully find rewards. Using optogenetic inactivation on a randomized subset of trials we found a deficit in the animal's ability to use landmarks to guide localizing rewards (*Figure 1F*). This could potentially be explained by a pure path integration deficit. However, if this is the case, RSC should exhibit a purely ego-centric representation of space, that is aligned to the start of a given trial. In contrast, we find that spatial representations in RSC are anchored by allocentric (landmark) cues and maintained by self-motion feedback after the animal has passed the visual cue (*Figure 3*). We note that in our task, trial onset is not an optimal reference point to test whether RSC acts as an internal odometer, represents sensory inputs, or, as we argue, integrates both. We found a degradation as a function of distance from the onset of the task. However, the onset of a trial is a poor reference point due to the randomized location at which the animal is placed on the track (50–150 cm before the landmark). Significantly more neurons are anchored by the landmark, compared to the trial onset. Therefore, what may look like a neural code that gets noisier as a function of distance from the trial start, may indeed only reflect the fact that landmarks are at varying distances from the trial start in a given session. Therefore, the landmark is a better reference point to test the accumulation of noise is therefore the landmark. In *Figure 3G*, we show that population vector cross correlation decreases slowly after the landmark with an uptick shortly before the animal gets to the reward zone. While this is not an optimal measure, and indeed this task was not designed to test error accumulation, we believe that this is evidence for error accumulation in VR.

Numerous experimental and theoretical studies have emphasized the importance of landmarks for anchoring spatially tuned cells during navigation (*Burak and Fiete, 2009*; *Campbell et al., 2018*; *Funamizu et al., 2016*; *Gothard et al., 1996a*; *Harvey et al., 2012*; *Jeffery, 1998*; *Knierim et al., 1998*; *Muller and Kubie, 1987*), yet the mechanisms that combine inputs from different modalities to represent landmarks remain poorly understood. We found that simple linear summation of visual and motor inputs was insufficient to explain landmark encoding in RSC. Instead, a nonlinear mechanism (or multiple mechanisms) underlie the integration of these variables to produce robust visuospatial responses during navigation. Active navigation (in virtual reality) sharpened spatial tuning and increased robustness of encoding in RSC compared to viewing the movie without being engaged in the task. Interestingly, the amplitude of recorded transients was unchanged, suggesting the presence of a thresholding process in the circuit (*Figure 4I–L*). Locomotion during decoupled stimulus presentation significantly increased the robustness of encoding while having very little effect on the fidelity of spatial tuning or transient amplitude. This indicates that motor input broadly pushes neurons towards spiking but does not contribute to its spatial tuning (*Figure 5F–I*). Furthermore, our data show that visual input alone is insufficient to explain the fidelity of spatial tuning we observed during virtual navigation (*Figure 5D*). Our results indicate that the most likely mechanism underlying supralinear integration in RSC is the multiplicative effects of significantly improved fidelity of spatial tuning and increased likelihood of emitting transients within a neurons receptive field. While the latter seems to be mediated by motor inputs, the nature and source of an anchoring signal is unclear but may originate in the hippocampal formation where landmarks have been found to sharpen spatial tuning of neurons (*Campbell et al., 2018*; *Gothard et al., 1996a*; *Knierim et al., 1995*). Animals developed a more robust code for space during task acquisition (*Figure 6E and F*), suggesting that this supralinear mechanism may be the result of learning the spatial significance of a visual landmark cue. Understanding how learning shapes the integrative properties of individual neurons is an exciting avenue for future studies.

The spatial tuning we observed in task active neurons in dysgranular RSC appears similar to those of place cells (see *Figure 2B* and *Figure 3A and D*). This is consistent with findings in *Mao et al. (2017)*, who report that spatial tuning in RSC is somewhat degraded when tactile or visual cues are removed from a belted treadmill. However, RSC may not exhibit spatial representations that differ from CA1 when sensory information regarding goals is absent. Consistent with this, we find that RSC is strongly biased to encode behaviorally relevant visual cues that inform the animal about the location of a reward. This robust code only emerges after learning the spatial significance of the visual cues (*Figure 6*). These findings are complementary to previous studies showing that RSC

conjunctively encodes information in egocentric and allocentric reference frames (*Alexander and Nitz, 2015*; *Alexander and Nitz, 2017*) as well as other variables (*Smith et al., 2012*; *Vedder et al., 2017*).

RSC's bias to encode behaviorally relevant stimuli is particularly interesting in light of its relationship with axonal inputs from V1 (*Figure 7*). Using the same landmark-dependent navigation task, we found that V1 axons exhibited comparable receptive field tunings as RSC neurons. However, these responses were substantially less modulated by task engagement (*Figure 7I*), suggesting that V1 axons encode visual features more faithfully. Previous studies have shown that neurons in V1 are themselves locomotion modulated (*Niell and Stryker, 2010*; *Saleem et al., 2013*). We report less locomotion modulation in V1 compared to RSC. The exact nature of this difference may be either of a qualitative nature, in which individual neurons are impacted differently by locomotion, or quantitative, in which across the population fewer neurons are locomotion modulated in V1 compared to RSC. We cannot disambiguate between these possibilities, as the overall fraction of locomotion-modulated neurons in V1 has not yet been established. Furthermore, it is not clear if only a specific functional subset projects to RSC, which may be more or less locomotion-modulated relative to the rest of V1.

The modulation we did observe may be the result of strong top-down inputs from RSC itself (*Makino and Komiyama, 2015*) or from other regions (*Zhang et al., 2014*). This is congruent with a recent study showing that activity in RSC is more correlated with V1 during locomotion compared to quiescent periods (*Clancy et al., 2019*). Indeed, before learning the behavioral significance of visual features in a novel environment, RSC may initially receive purely visual inputs from V1. As the animal learns to navigate in the new environment, feedback from RSC to V1 (as well as other areas such as ACC) may lead to modulated responses based on behavioral relevance in primary visual cortex, as reported by an increasing number of studies (*Attinger et al., 2017*; *Pakan et al., 2018*; *Poort et al., 2015*; *Saleem et al., 2018*). Our data provide evidence that RSC may act as a critical processing node that gates behaviorally relevant visual inputs and relays them to the entorhinal cortex and other areas involved in spatial navigation, where its readout may be used to anchor spatially tuned neurons such as grid cells (*Burak and Fiete, 2009*; *Campbell et al., 2018*) or head-direction cells (*Jacob et al., 2017*). While RSC receives inputs from V2, M2, and other cortical areas (*Oh et al., 2014*; *Sugar et al., 2011*), functional imaging studies show RSC to be uniquely engaged during landmark-based navigation (*Auger et al., 2012*; *Epstein, 2008*; *Epstein and Vass, 2014*; *Maguire, 2001*; *Vann et al., 2009*), suggesting RSC is indeed a key locus for integrating visual and spatial information compared to other association areas.

Finally, this work provides novel insights into the neural mechanisms underlying cognitive computations. Our results are consistent with data from humans with RSC lesions who show an impaired ability to use environmental cues for navigation, as well as neuroimaging studies that show increased activity in RSC during spatial behaviors (*Cho and Sharp, 2001*; *Ino et al., 2007*; *Julian et al., 2018*; *Maguire, 2001*; *Robertson et al., 2016*; *Vann et al., 2009*). Leveraging RSC to unravel how multiple input streams are integrated during higher level associative processes like navigation may in the future provide novel insights into the mechanisms of cognition and its dysfunction in Alzheimer's disease and other currently intractable brain disorders.

## Materials and methods

### Animals and surgeries

All animal procedures were carried out in accordance with NIH and Massachusetts Institute of Technology Committee on Animal care guidelines. Male and female mice were singly housed on a 12/12 hr (lights on at 7 am) cycle.

C57BL/6 mice (RRID: IMSR_JAX:000664) were implanted with a cranial window and headpost at 7–10 weeks of age. First, the dorsal surface of the skull was exposed and cleaned of residual connective tissue. This was followed by a 3 mm wide round craniotomy centered approximately 2.5 mm caudal of the bregma. To minimize bleeding, particularly from the central sinus, the skull was thinned along the midline until it could be removed in two pieces. AAV1.Syn.GCaMP6f.WPRE.SV40 was injected at 2–6 injection sites, 350–600 μm lateral of the midline in boluses of 50–100 nl at a slow injection rate (max. 50 nl/min) to prevent tissue damage. Following injections, a cranial window was

placed over the craniotomy and fixed with cyanoacrylate glue (Krazy Glue, High Point, NC, USA). The windows consisted of two 3 mm diameter windows and one 5 mm diameter window stacked on top of each other (Warner instruments CS-3R and CS-5R, Hamden, CT, USA). The windows were glued together with optical glue (Norland Optical Adhesive #71, Edmund Optics, Barrington, NJ, USA). Cranial windows consisted of 3 (instead of 2) stacked windows to account for increased bone thickness around the midline and minimize brain motion during behavior. Subsequently, the head-plate was attached using cyanoacrylate glue and Meatbond (Parkell Inc NY, USA) mixed with black ink to avoid light leaking into the objective during recordings.

Mice prepared for imaging of V1 boutons in RSC had GCaMP6f injected into V1 (~2.49 mm lateral, 3.57 caudal) through small burr holes at a depth of 600 µm to target primarily layer 5 neurons. For the rest of the procedure, the same steps as for imaging of RSC neurons were followed.

For the optogenetic inactivation during behavior experiment, VGAT-Ires-Cre knock-in mice (VGAT is encoded by *Slc32a1*, RRID: IMSR_JAX:028862) on a C57BL/6 background (The Jackson Laboratory) were injected with flexed channelrhodopsin-2 (ChR2, AAV5.ef1a.DIO.ChR2.eYFP, University of Pennsylvania Vector Core) in 2–3 locations along the AP axis of RSC (50–100 nl per injection). Prior to injection the location of the central sinus was identified by placing saline on the skull and waiting until it was translucent. This was done because the overlying sagittal suture can be inaccurate in identifying the midline of the brain. One ferrule was placed centrally on each hemisphere over RSC. Each ferrule was calibrated prior to implantation to ensure the same light intensity was provided into each hemisphere.

## Virtual reality setup

Head-fixed mice were trained to run down a virtual linear corridor by locomoting on a polystyrene cylinder measuring 8 cm in width and 20 cm in diameter (Graham Sweet Studios, Cardiff, UK). The cylinder was attached to a stainless-steel axle mounted on a low-friction ball bearing (McMaster-Carr #8828T112, Princeton, NJ, USA). Angular displacement of the treadmill was recorded with an optical encoder (US Digital E6-2500, Vancouver, WA, USA). A custom designed head-restraint system was placed such that animals were comfortably located on the apex of the treadmill. Rewards were provided through a lick spout (Harvard Apparatus #598636) placed within reaching distance of the mice's mouth. Timing and amount were controlled using a pinch valve (NResearch 225PNC1-21, West Caldwell, NJ, USA). Licking behavior was recorded using a capacitive touch sensor (SparkFun #AT42QT1010, CO, USA) connected to the lick spout. The virtual environment was created and rendered in real time in Matlab using the software package ViRMeN (*Aronov and Tank, 2014*) as well as custom written code. Two 23.8' computer screens (U2414H, Dell, TX, USA) were placed in a wedge configuration to cover the majority of the mice's field of view.

## Behavioral task design and training

After mice had undergone preparatory surgery, they were given at least one week to recover before water scheduling began. Initially, mice received 3 ml of water per day in the form of 3 g of HydroGel (ClearH$_2$O, Watertown, MA, USA), which was gradually reduced to 1.0–1.5 g per day. During this period, mice were handled by experimenters and habituated to being head restrained as well as running on a cue-less version of the virtual corridor. During habituation, mice were given small water rewards to allow them to acclimate to receiving 10% sugar-water rewards through a spout during head-restraint. Behavioral training began once mice were locomoting comfortably, as assessed by posture and gait. Initially, mice were trained on one trial type alone (short track). Each trial started at a randomized distance from the landmark (150–50 cm, drawn from a uniform distribution). The wall pattern consisted of a uniform pattern of black dots against a dark gray background to provide generic optic flow information. The view-distance down the corridor was not limited.

The landmark cues were 40 cm wide and extended above the walls of the corridor (see *Figure 1B*). After passing the landmark, mice were able to trigger rewards by licking a fixed distance from the landmark. The reward zone was 20 cm long but not indicated in any way so that the animals had to use self-motion cues and the location of the landmark to locate it. If an animal passed through the reward zone without licking, an automatic 'reminder' reward was dispensed. Each reward bolus consisted of 4–6 µl of 10% sucrose water. Sucrose was added to maximize training success (*Guo et al., 2014*). Reward delivery marked the end of a trial and animals were 'teleported' into

a 'black box' for at least 3 s. In some training and recording sessions, animals were required to not lick or run for 3 s, however that requirement was later removed. Training using only one trial type was carried out daily in 30 to 60 min sessions until licking behavior was reliably constrained to after the landmark. At that point, the second trial type (long track) was introduced. Training using two tracks was carried out until the licking behavior of mice indicated that they used landmark information to locate the reward ('experts', typically 2–4 weeks). An empirical bootstrap shuffle test was used to calculate confidence intervals and evaluate whether or not mean first lick locations where significantly different. At that point, mice were transferred to the 2-photon imaging rig. In some instances, a small number of training sessions with the recording hardware running were carried out on the imaging setup to acclimatize animals.

The spatial modulation z-score (SMZ) was calculated by randomly rotating the location of licks within each trial by a random amount. The fraction of correctly triggered trials within this new, shuffled session was calculated by evaluating whether at least one lick was within the rewarded zone. This process was repeated 1000 times and a null distribution of fraction successful from random licking was calculated.

## Optogenetic inactivation experiment

Optogenetic inactivation was carried out on mice that had been trained to expert level. Once they reached proficiency at using landmarks to locate rewards, the masking light was introduced. Animals were allowed a small number of sessions to habituate to the masking light before inactivation trials were introduced. The masking stimulus was provided by two 465 nm wavelength LEDs mounted on top of the computer screens facing the animal (Thorlabs LED465E, Thorlabs, NJ, USA). Optogenetic stimulation light was provided by a 470 nm fiber coupled LED (Thorlabs M470F3) powered by a Cyclops LED driver (*Newman et al., 2015*). Stimulation consisted of a solid light pulse with a maximum duration of 10 s (*Lewis et al., 2015*). Stimulation was provided on half of the trials in a random order, with the only exception that no two consecutive trials could be stimulation trials. Light intensity ranged from 1 to 10 mW and was calibrated individually for each animal. Each animal was observed during stimulation trials and checked for no visible effects on behavior such as change in posture or gait. No difference was found in mean running speed or licks per trial when the stimulation light on compared to when only the masking stimulus was shown (*Figure 1—figure supplement 1*). The task scores on mask only trials were compared to the task scores on mask + stimulation trials to assess deficits in the mice's ability to use the landmark as a cue to locate rewards.

## Two-photon imaging

A Neurolabware 2-photon microscope coupled to a SpectraPhysics Insight DeepSee II were used for GCaMP6f imaging. To prevent photodamage or bleaching during extended recording periods, a 4x pulse splitter was placed in the light path (*Ji et al., 2008*). The virtual reality software ran on a separate computer that was connected to the image acquisition system. Start and end of recording sessions were controlled by the virtual reality software to ensure synchrony of behavior and imaging data. Animals were placed in the head restraint and had a custom-designed 3D printed opaque sleeve placed over their cranial window to block light from the VR screens from leaking into the objective. The scope was lowered and suitable FOV identified before recordings began. Neurons in RSC were recorded at a wavelength of 980 nm. During V1 bouton recordings, the wavelength was switched to 920 nm. This was done to minimize autofluorescence from the dura mater, which is more pronounced at 980 nm excitation, especially during superficial recordings. Images were acquired at a rate of either 15.5 Hz or 31 Hz. In a subset of recordings, an electronically tunable lens was used to record from multiple FOVs in the same animal and session. In all but one cases, dual-plane imaging was used at a rate of 31 Hz, resulting in 15.5 Hz per plane acquisition. In a single recording session, six planes were acquired at 5.1 Hz. The two planes with most somas where included in this study. Recordings were acquired continuously throughout each session as opposed to epoch-based on trials.

## Image processing, segmentation, and signal extraction

Custom written Mathworks Matlab code was used for image registration, segmentation and signal extraction. Each recording session was stabilized using an FFT based rigid algorithm to register each

frame to a template created from a subset of frames drawn at random from the whole session. This was followed by creating a pixel-by-pixel local cross-correlation and global cross-correlation maps. Regions of interest were drawn semi-automatically based on local cross-correlation from an experimenter defined seed-point. In addition to cross-correlation, global PCA, mean intensity, and other maps were created to aid identification of neurons and axonal boutons. Since the FOV was the same for virtual navigation and decoupled stimulus presentation, the same ROI maps created during virtual navigation could be used for decoupled stimulus presentation. During signal extraction, the mean brightness value of all pixels within a single ROI was calculated. A neuropil 'donut' was automatically generated around each ROI to allow for subtraction of local brightness from ROI signal. $\Delta F/F$ was calculated using a 60 s sliding time window. For neurons, $F_0$ was calculated from the bottom $5^{th}$ percentile of data points within the sliding window. For boutons, the bottom $50^{th}$ percentile was used to calculate $F_0$. Neuropil signal was subtracted from ROI signal prior to calculating $\Delta F/F$. Each ROI time course was manually inspected prior to inclusion into analysis. ROIs were excluded if they had few transients (<0.5/min). Transients were identified as detected whenever the $\Delta F/F$ signal was above six standard deviations for at least 500 ms. The ROI time course was then aligned and resampled to match behavioral data frame-by-frame using custom code and the Scipy signal processing toolbox (*Jones et al., 2001*). To test for long term imaging side effects despite using a pulse splitter, we tested for baseline drift of mean frame brightness for each included recording session (*Figure 4—figure supplement 2*).

Experiments in which the same neurons were recorded in naïve and expert animals, field of views (FOV) were matched manually at recording time. For signal extraction, ROIs drawn on the naïve FOV were transferred to the expert FOV and, where necessary, manually adjusted overlay on the same neuron.

## Neuron and axonal bouton classification

The time course of each neuron was split into individual trials and aligned to one of three anchor points: trial onset, landmark, and reward. For the neuron to be considered task engaged it had to fulfill the following criteria: 1) $\Delta F/F$ had to exceed three standard deviations of the ROI's overall activity on at least 25% of all trials; 2) the mean $\Delta F/F$ across trials had to exceed a peak z-score of 3 at its peak. The z-score for each ROI was determined by randomly rotating its $\Delta F/F$ time course with respect to its behavior 500 times and the peak value of the mean trace was then used to calculate the peak z-score; 3) the minimum of the mean trace amplitude (i.e. highest – lowest value) had to exceed 0.2 $\Delta F/F$. Criteria for axonal boutons were the same with the exception that the minimum mean trace amplitude was 0.1 $\Delta F/F$. For the neurons that passed these criteria, the alignment point that resulted in the largest mean response was determined. To avoid edge-cases at the beginning and end of the track, the mean trace was only calculated for bins where at least 50% of trials were present. For the comparison of peak amplitude in *Figure 4C and D*, the peak amplitude as a function of space, rather than time, was used. The landmark selectivity index was calculated for all neurons that were classified as landmark aligned on at least one trial type as LMI = $(LM_{short} - LM_{long})/(LM_{short} + LM_{long})$, where $LM_x$ refers to the peak response to the respective landmark. Only neurons that were classified as landmark-aligned neurons were included in that analysis. The fraction of neurons classified as trial onset, landmark, or reward were calculated from the total number of neurons with a baseline activity of at least 0.5 transients/min.

## Template matching decoder

A template matching decoder was used to assess the accuracy by which the trial type could be identified based on the activity of the different categories of neurons (trial onset, landmark, reward). First, template vectors were constructed for each trial type by calculating the mean response across trials within a session. The responses of the same neurons in individual trials were then compared to the template vectors, resulting in a similarity index for each trial type:

$$\Theta_\theta = \frac{\Sigma_{i=1}^{N} R_i^t R_i^\theta}{\left\|\boldsymbol{R^t}\right\| \cdot \left\|\boldsymbol{R^\theta}\right\|}$$

Here, $\Theta$ is the similarity index for trial type $\theta$ (short or long). $\boldsymbol{R^\theta}$ is the template vector for trial type $\theta$, $\boldsymbol{R^t}$ is a vector of the responses of all neurons in a given trial, and i...N are the indices of all neurons

of a given category. Whichever similarity index was higher for a given trial was considered the decoded trial type and compared to the trial type the animal was actually on.

## Population plot and population vector analysis

Population plots were created by binning the activity of each neuron as a function of space. Each bin was 5 cm wide and all data points falling within a bin were averaged to calculate the mean activity at that location. The first bin started at 100 cm distance from the landmark such that it contained data from at least 50% of trials on average. The activity in each bin was normalized to the bin with peak activity of the same neuron such that all data ranged from 0 to 1. To plot the mean activity of all neurons in this study, the data was split. Half the trials were randomly drawn to calculate the bin with peak activity. The other half of the trials was used to calculate activity to be plotted. The population vector cross-correlation was calculated similarly by randomly drawing half of the trials to construct one vector and using the other 50% trials to construct the other vector. For each spatial bin, the Pearson correlation coefficient was calculated. The location reconstruction error was calculated as the distance between the spatial bin with the highest cross-correlation value to the animals' actual location. As randomly splitting trials into halves can lead to spurious cross-correlation maps, this process was repeated 10 times and the mean cross-correlation coefficient and position reconstruction error for each spatial bin was calculated.

## Activity during decoupled stimulus presentation

The peak response during decoupled stimulus presentation was evaluated by aligning each neuron or bouton to its preferred alignment point during virtual navigation. The response was then measured at the same point relative to that alignment point (in space) where it showed its peak response during virtual navigation. To allow for small shifts in peak activity during decoupled stimulus presentation, a window of $\pm$ 20 cm was introduced and the peak value within that window was used for analysis. Transients were identified whenever $\Delta F/F$ in a given trial (binned into 2 cm spatial bins) rose above six standard deviations of that neurons baseline activity (70th percentile of data points) for at least two consecutive bins. Transients located outside $\pm$ 60 cm of the mean peak during virtual navigation were excluded. Jitter was calculated as the standard error of the difference between mean peak and transient peak locations.

## Motor vs. no motor analysis

The effect of concurrent motor and visual inputs during decoupled stimulus presentation was assessed by grouping trials based on whether the animal was running or stationary. A trial was considered a 'running' trial if its average speed exceeded 3 cm/sec in a $\pm$ 50 cm time window around its peak response relative to the landmark during virtual navigation. Only neurons with at least three running and non-running trials were included. To allow for slight mismatches between a neuron's peak response during virtual navigation and decoupled stimulus presentation, the peak within 20 cm of the VR peak was used. Activity in the black box was calculated as a neuron's response 1.5 s after onset of showing black screens with a movement time window of $\pm$1 s. The relative response amplitudes were calculated by normalizing the activity of each neuron to its activity during virtual navigation. Only sessions in which a given animal ran and was stationary were included in this analysis.

Running speed modulation analysis was carried out by calculating the average running speed during the transient (e.g. if the transient was 500 ms long, the average running speed during those 500 ms was calculated) and correlating it to the peak amplitude of the transient. Only neurons where at least 10 transients where present on the respective trial type were included in this analysis.

## Laminar analysis

To assess differences in neuronal responses of superficial vs deep neurons in RSC, the depth of the recordings was used to determine which layer neurons belonged to. RSC does not possess a layer 4, and layers 2/3 and 5 are separated by a section of relatively few cell bodies. In addition, layer five is comparatively superficial, starting at only 300 µm below the pia (*Lein et al., 2007*). This made identification of cortical layers during in vivo 2-photon imaging possible. We split recordings into layer 2/3 and layer five recordings based on depth below the pia. In one recording, a Rbp4-Cre positive animal, which expresses Cre in many layer 5 cells, was used in conjunction with a flexed GCaMP6f

construct. The recording depth for this animal was congruent with other recordings in which we located layer five based on recording depth alone.

## Acknowledgements

We are grateful to Ben Scott, Jeff Gauthier, Elias Issa, Matt Wilson, and Ashok Litwin-Kumar for comments on the manuscript. We thank Enrique Toloza, Jakob Voigts, and the rest of the Harnett lab for critical feedback on the project, analyses, and manuscript. Support was provided by NIH RO1NS106031, the McGovern Institute for Brain Research, the NEC Corporation Fund for Research in Computers and Communications at MIT, and the Klingenstein-Simons Fellowship in Neuroscience (to MTH).

## Additional information

### Funding

| Funder | Grant reference number | Author |
| --- | --- | --- |
| National Institutes of Health | RO1NS106031 | Mark Thomas Harnett |
| Esther A. and Joseph Klingenstein Fund | | Mark Thomas Harnett |
| NEC Corporation Fund for Research in Computers and Communications, MIT | | Mark Thomas Harnett |

The funders had no role in study design, data collection and interpretation, or the decision to submit the work for publication.

### Author contributions

Lukas F Fischer, Conceptualization, Data curation, Software, Formal analysis, Validation, Investigation, Visualization, Methodology, Project administration; Raul Mojica Soto-Albors, Formal analysis, Investigation; Friederike Buck, Conceptualization, Investigation; Mark T Harnett, Conceptualization, Supervision, Funding acquisition

### Author ORCIDs

Lukas F Fischer https://orcid.org/0000-0001-9422-3798
Raul Mojica Soto-Albors https://orcid.org/0000-0002-6987-5417
Mark T Harnett https://orcid.org/0000-0002-5301-1139

### Ethics

Animal experimentation: All animal procedures were carried out in accordance with NIH and Massachusetts Institute of Technology Committee on Animal care guidelines (CAC #0518-040-21).

### Decision letter and Author response

Decision letter https://doi.org/10.7554/eLife.51458.sa1
Author response https://doi.org/10.7554/eLife.51458.sa2

## Additional files

### Supplementary files

- Source code 1. Analysis code.

- Transparent reporting form

### Data availability

Behavior and imaging data available on Dryad.

The following datasets were generated:

| Author(s) | Year | Dataset title | Dataset URL | Database and Identifier |
|---|---|---|---|---|
| Lukas F, Raul MS, Friederike B, Mark Thomas H | 2019 | Data from: Representation of Visual Landmarks in Retrosplenial Cortex | https://doi.org/10.5061/dryad.6vd388v | Dryad Digital Repository, 10.5061/dryad.6vd388v |
| Lukas F, Raul MS, Friederike B, Mark Thomas H | 2019 | Representation of visual landmarks in retrosplenial cortex | http://doi.org/10.5061/dryad.8gtht76k8 | Dryad Digital Repository, 10.5061/dryad.8gtht76k8 |

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
