## [Decision Letter]

**Acceptance summary:**

This paper features one of the first two-photon imaging experiment in a complex cognitive task in rodents causally involving an insufficiently characterized associative area, the retrosplenial cortex. It elegantly demonstrates with two-photon calcium imaging recordings in naive and trained animals (even in some cases following the same neurons through the course of learning) that a representation of the task, linking the visual cues to the task parameters, emerges with learning in retrosplenial cortex. These findings will impact a broad range of fields of study, including network mechanisms of learning.

**Decision letter after peer review:**

Thank you for submitting your article "Representation of visual landmarks in retrosplenial cortex" for consideration by *eLife*. Your article has been reviewed by three peer reviewers, and the evaluation has been overseen by a Reviewing Editor and Laura Colgin as the Senior Editor. The reviewers have opted to remain anonymous.

The reviewers have discussed the reviews with one another and the Reviewing Editor has drafted this decision to help you prepare a revised submission.

Summary:

Harnett and colleagues are studying coding for visual landmarks in a virtual reality task where different visual landmarks predict reward at different distances. They show that retrosplenial cortex inactivation worsens behavior. Using two-photon calcium imaging they show neurons whose firing is locked to running onset, visual landmark, or reward. The fidelity of firing decreases when visual stimuli are decoupled from behavior. V1 axons in retrosplenial cortex had similar responses but were less modulated by task engagement.

This is a very interesting study with novel results for the community. The experiments are performed carefully, for the most part, and the analysis is thorough. However, the manuscript is not always clear. Some issues could be better discussed, and the focus on navigation should be diminished, as the task, and the associated role of RSC, may very well be restricted to visual discrimination associated to path length estimation.

Essential revisions:

1) The authors have used genetically-modified mice to specifically target GABAergic cells optogenetically. But it is not clear how much of RSC is being manipulated by this approach. Authors say multiple injections were made along the rostro-caudal axis, but there is no histology. This does not make clear how much of the cortex would be covered by the laser. Some kind of additional corroboration would be helpful to see what extent of RSC is affected.

2) Regarding the task, it is important to carefully clarify where animals use landmarks to localize themselves in space or whether they have some sort of visually cued path integration strategy. This raises the question of whether, during RSC inactivation, spatial navigation is impaired or mice are unable to recognize the two landmarks/to associate a landmark with the correct distance to perform. The task score used by the authors does not distinguish between these possibilities. One way to answer this question is to more carefully analyze the mouse behavior. For example, when do animals stop during inactivation trials? Do they allocate short and long paths to the incorrect landmark? Or do they behave randomly? Do speed profiles change? Finally, it is important to discuss these two possibilities and to modify the statements in the manuscript about the impact of RSC on 'navigation' as the effect could be only on 'visual recognition'.

3) The authors showed in Figure 3 that, if the activity is aligned to the onset, this results in a very noisy coding. According to the authors, this result demonstrates that RSC activity is independent of an internal odometer that allows computing distances from a starting position. But, if RSC activity reflects an internal (i.e. self-motion based) odometer, one expects noisier activity at distances farther from the onset (due to error accumulation). From Figure 3, it seems that firing selectivity starts to be lost at around 30cm from the onset, for both short and long. The authors should discuss if this is compatible with error accumulation in virtual reality.

4) As a control for task-dependent landmark selectivity, the authors performed a decoupled stimulus presentation. In addition, trials in which mice were immobile and trials in which they moved were dissociated. The authors must clarify how these experiments were performed. Did the authors perform passive viewing and active locomotion independently? If this were not the case, how was optic flow controlled with respect to spontaneous movements? Please provide more precise explanations, and discuss the impact of spontaneous movements on the results.

5) The authors found a drastic modification in cell activity during the decoupled condition, particularly when animals were immobile. What happened during virtual navigation when animals stopped at the receptive field? From the corresponding figure (Figure 5B), it seems that in the decoupled experiment when the mouse was not moving the cell was silent, but when the mouse was moving the cell had the same receptive field but with lower activity. Can the effect seen during task engagement be explained by a modulation in animal movement speed? Please clarify this point in particular in the linearity analysis for visual responses and motor context. On the same line, the authors make the claim that, "This indicates that encoding of behaviourally-relevant variables in RSC is dependent on ongoing behaviour". This statement should be mitigated, with respect to the analysis above.

6) How much do animals pay attention to landmarks? Is there any possibility that cell activity is modulated by this parameter? Is it possible that in the decoupled condition mice do not pay attention to the landmarks because they are irrelevant, thus reducing cell responses? Please discuss and ideally provide pupil dilation data.

7) The authors only show cell responses once the animals are well-trained on the task and then use a decoupling approach to show that the responses are task/navigation sensitive. It would be more convincing to show that the pattern of activity changes with task acquisition (i.e., changes in individual cell responses, changes in number of cells responding, or changes in the ensembles that are activated) to show that this isn't simply a stimulus response. A comparison with the first trials where the animals are seeing the cues and running without knowledge of rewards or reward locations would disambiguate this.

8) Overall number of cells responding does not necessarily equate to a greater role in a particular component of the task. What is more informative is how well the cell responses correlate with what the animals have learned in terms of distance prediction. And, patterns of cell activity may be more informative than absolute numbers. The authors should quantify this further (for example, using population activity).

9) Regarding laminar differences in landmark coding, the justification is unclear for this. There is no rationale as to what differences would be expected in terms of the connectivity of the different layers. The authors write "….L2/3 may be key for encoding landmarks during ongoing behaviour". One may assume this is because there is a greater proportion of cells engaged, but greater overall numbers does not necessarily a greater role – it may be more informative to model how networks within RSC support behavior i.e., interactions between the layers.

10) The present study appears to be focused on dysgranular RSC; the main RSC subregions are known to have different connectivity and responsiveness, so the authors should be more specific in terms of which part of RSC they are referring to. Moreover, the authors distinguished different RSC layers but did not evaluate possible differences between granular and dysgranular parts of the RSC. Given that these two areas contain functionally different head-direction cell categories (Jacob et al., 2017), is there any possibilities that they respond differently to landmarks? This should be clarified.

---

## [Author Response]

Essential revisions:1) The authors have used genetically-modified mice to specifically target GABAergic cells optogenetically. But it is not clear how much of RSC is being manipulated by this approach. Authors say multiple injections were made along the rostro-caudal axis, but there is no histology. This does not make clear how much of the cortex would be covered by the laser. Some kind of additional corroboration would be helpful to see what extent of RSC is affected.

The reviewers’ raise an important point. We have now included confocal images of an animal injected with the same injection protocol as the animals used in the inactivation experiment (Figure 1—figure supplement 1N). This experiment shows ChR2 expression along the rostro-caudal extent of RSC. The ChR2 virus used in the study is conjugated with an eYFP marker which has been amplified using immunohistochemistry. This is evidence that many GABAergic neurons are transfected by ChR2 throughout the majority of RSC A30. We point out that in the optogenetic experiments we placed optical ferrules on top of the skull above each hemisphere, which increased the spread of the light by utilizing the skull as a diffuser. Therefore, most excitatory neurons in A30 are likely inactivated during the optogenetic stimulation in our experiment.

2) Regarding the task, it is important to carefully clarify where animals use landmarks to localize themselves in space or whether they have some sort of visually cued path integration strategy. This raises the question of whether, during RSC inactivation, spatial navigation is impaired or mice are unable to recognize the two landmarks/to associate a landmark with the correct distance to perform. The task score used by the authors does not distinguish between these possibilities. One way to answer this question is to more carefully analyze the mouse behavior. For example, when do animals stop during inactivation trials? Do they allocate short and long paths to the incorrect landmark? Or do they behave randomly? Do speed profiles change? Finally, it is important to discuss these two possibilities and to modify the statements in the manuscript about the impact of RSC on 'navigation' as the effect could be only on 'visual recognition'.

Recognizing landmarks as such (i.e. understanding and recalling the navigational meaning of a given visual cue in the environment) versus associating a visual feature with a given path to a reward may both be functions of RSC. Our study was not designed to disambiguate these two possibilities. However, the revised manuscript now contains additional analyses, figures, and discussion to describe the specific behavioral changes animals exhibit on RSC inactivation trials. First, we looked at running speed profiles and found no significant difference between stimulation trials vs. mask only trials (Figure 1—figure supplement 1O,P), suggesting that locomotion behavior was not significantly affected by RSC inactivation. Second, we did a label-shuffle test (randomly switching the labels for long and short trials) and re-analyzed licking behavior. If animals were unable to discriminate, or even see, landmarks, then the task score resulting from a label-shuffle test should not be different from the task scores on stimulation trials. Instead, we see a decrease in task score, suggesting that they have diminished ability to locate rewards based on the landmark. These new analyses now provide a more precise evaluation of the behavioral deficits inactivation of RSC causes.

In addition, we are submitting with this rebuttal two separate experiments we used to test whether inactivating of RSC affects mouse vision or motor behavior. We used two simple derivative versions of our landmark navigation task: one required animals to discriminate between two visual cues and the other required them to locomote two different distances to visually cued reward locations (see Author response image 1). In both experiments, no effect of RSC inactivation (via muscimol) could be seen compared to sham injections. We acknowledge the different nature of muscimol inactivation compared to optogenetic inactivation: we provide these results in the rebuttal to corroborate the point that our optogenetic results are not a result of RSC-mediated visual or motor function impairment.

**Author response image 1. respfig1:** Muscimol inactivation on visual discrimination and motor control tasks. (**A**) Schematic of visual discrimination task (top) in which reward was available at one visual cuebut not the other. The distance to both visual cues was equal. Animals learned to discriminate reliably (middle raster plot) and task performance was not affected by multi-site (2-4) bilateral muscimol injections into RSC A30 (bottom raster plot). (**B**) Behavioral D-Prime showing that all animals learned the task in ~10-12 sessions (n=4 mice). (**C**) D-Prime during sham and muscimol injections. (**D**) Top: schematic of motor control experiment where animals ran different distances to obtain rewards at either visual cue. On the short trials, mice had to travel an average of 140 cm while on the long trials they had to travel 200 cm on average. Both visual cues were rewarded. Raster plots show behavior during sham and muscimol multisite bilateral injection into RSC A30. (E,G) Running speed profiles and mean running speeds during sham and muscimol injection. (**F**) Task score during sham and muscimol injection.

3) The authors showed in Figure 3 that, if the activity is aligned to the onset, this results in a very noisy coding. According to the authors, this result demonstrates that RSC activity is independent of an internal odometer that allows computing distances from a starting position. But, if RSC activity reflects an internal (i.e. self-motion based) odometer, one expects noisier activity at distances farther from the onset (due to error accumulation). From Figure 3, it seems that firing selectivity starts to be lost at around 30cm from the onset, for both short and long. The authors should discuss if this is compatible with error accumulation in virtual reality.

We thank the reviewers for this constructive challenge of our interpretation of the data. We do not feel our data can be used to strongly argue either for or against error accumulation in VR. Aligning neurons by trial onset, as we did, can produce apparently “very noisy coding” because animals start each trial at randomized locations with respect to the landmark (50-150 cm). Significantly more neurons are anchored by the landmarks compared to the trial onset (as shown in Figure 2), which may well account for the “noisy coding” in Figure 3. We have added a discussion of this important point to the main text (Discussion paragraph two).

4) As a control for task-dependent landmark selectivity, the authors performed a decoupled stimulus presentation. In addition, trials in which mice were immobile and trials in which they moved were dissociated. The authors must clarify how these experiments were performed. Did the authors perform passive viewing and active locomotion independently? If this were not the case, how was optic flow controlled with respect to spontaneous movements? Please provide more precise explanations, and discuss the impact of spontaneous movements on the results.

We have expanded and clarified our description of the decoupled experiments in the Results section. We now explicitly state that data from immobile and mobile periods were all collected during the same sessions (subsection “Nonlinear integration of visual and motor inputs in RSC”). Trials within a session were separated based on whether the animal locomoted as the virtual environment passed a neuron’s receptive field (Figure 5A and 5B, ‘+ motor’ trials, running speed > 3 cm/sec in a ± 50 cm window). ‘No input’ and ‘motor only’ conditions were measured while the animal was in the blackbox in between trials using similar criteria as before but with a 1.5 second time window (instead of a spatial window) for locomotion. Thus, any brief spontaneous movements that may have occurred within this window, but were below threshold, were labelled passive viewing. Despite the possibility that these small movements could impact neural responses, we nonetheless observed a striking contrast between locomoting and passively viewing animals. During decoupled stimulus presentation, optic flow was set at a constant speed and therefore not controlled with regards to spontaneous movement. The time window sizes were deliberately chosen to be generous so that we compared trials when the animal was consistently locomoting to periods of passive viewing.

5) The authors found a drastic modification in cell activity during the decoupled condition, particularly when animals were immobile. What happened during virtual navigation when animals stopped at the receptive field? From the corresponding figure (Figure 5B), it seems that in the decoupled experiment when the mouse was not moving the cell was silent, but when the mouse was moving the cell had the same receptive field but with lower activity. Can the effect seen during task engagement be explained by a modulation in animal movement speed? Please clarify this point in particular in the linearity analysis for visual responses and motor context. On the same line, the authors make the claim that, "This indicates that encoding of behaviourally-relevant variables in RSC is dependent on ongoing behaviour". This statement should be mitigated, with respect to the analysis above.

We have added further analyses and clarifications to the manuscript to address these points (subsection “V1 inputs to RSC represent task features but are less modulated by active navigation”). Specifically, Figure 6A and B now show running speed modulation analysis for every task active neuron. We also carried out a Pearson correlation analysis between the amplitude of each transient and the average running speed of that animal at the location of the transient. We found that only a small number of neurons that showed a correlation between transient amplitude and running speed (10.5% for short trials and 14.4% on long trials). However, those neurons whose correlations were significant featured both positive and negative correlations (Figure 4—figure supplement 1B), and therefore cannot explain the results we present in Figure 5. Expert mice typically did not stop along the track: they completed the vast majority of trials in one go. We therefore could not find a reasonable number of instances where animals stopped inside a given neuron’s receptive field in order to conduct the requested analysis. We modified the sentence suggested by the reviewer.

6) How much do animals pay attention to landmarks? Is there any possibility that cell activity is modulated by this parameter? Is it possible that in the decoupled condition mice do not pay attention to the landmarks because they are irrelevant, thus reducing cell responses? Please discuss and ideally provide pupil dilation data.

We have now added pupil data for 3 expert mice, which shows no difference in pupil dilation during active navigation and decoupled stimulus presentation (Figure 4—figure supplement 1F-H). Furthermore, we have added text regarding attentional issues during the decoupled stimulus presentation in the respective Results section (subsection “Active task execution sharpens spatial tuning and increases robustness of responses in RSC”).

7) The authors only show cell responses once the animals are well-trained on the task and then use a decoupling approach to show that the responses are task/navigation sensitive. It would be more convincing to show that the pattern of activity changes with task acquisition (i.e., changes in individual cell responses, changes in number of cells responding, or changes in the ensembles that are activated) to show that this isn't simply a stimulus response. A comparison with the first trials where the animals are seeing the cues and running without knowledge of rewards or reward locations would disambiguate this.

We agree that comparing neural activity before and after learning represents a powerful complement to our VR vs. uncoupled stimulus presentation dataset. We therefore performed a whole new set of experiments to obtain data from a new group of animals that were imaged on their first day of exposure to the VR task and again when they were experts (Figure 6, subsection “Active task execution sharpens spatial tuning and increases robustness of responses in RSC”). We matched fields-of-view and neuron ROIs (Figure 6A) across sessions. Congruent with our results in the uncoupled condition, we found that coding is substantially different before and after task acquisition (Figure 6B). Indeed, before learning, neuronal population activity showed a less accurate encoding of space compared with the same population of neurons after learning (Figure 6C-E). This provides evidence for our contention that the pattern of neural activity in RSC in expert animals is not simply stimulus driven.

We would like to note that before-and-after analysis of neuronal population activity in behaving mice that learn a complex task is exceedingly challenging. Few existing studies have followed the activity of the same neurons learning over days, let alone weeks. Those that have done so were conducted in primary sensory or motor cortex in animals learning relatively simple tasks (Huber et al., 2012; Pakan, Currie, Fischer, and Rochefort, 2018; Peron, Freeman, Iyer, Guo, and Svoboda, 2015; Peters, Chen, and Komiyama, 2014; Poort et al., 2015). We followed the same neurons over weeks in a task that required animals to learn the spatial relationships between different visual cues and reward locations and then use path integration to successfully complete trials. A particular strength of our study is establishing a complex behavioral paradigm that is associative cortex-dependent, which exploits a previously underutilized capability of 2-photon imaging during task acquisition.

8) Overall number of cells responding does not necessarily equate to a greater role in a particular component of the task. What is more informative is how well the cell responses correlate with what the animals have learned in terms of distance prediction. And, patterns of cell activity may be more informative than absolute numbers. The authors should quantify this further (for example, using population activity).

We agree that fractions of task-active neurons do not directly correspond to computational output. When we analyzed the correlation between the fraction of task active neurons (anchored by trial onset, landmark, or reward) and the respective task score (Figure 4—figure supplement 1E), we found no significant correlation. However, we found neurons that were differentially active within a session when we split the responses into the most and least accurate 25% of trials. A subset (21.6%) changed their activity by >0.5 ∆F/F. These changes were bidirectional: some neurons increased their activity while others decreased their activity based on how well an animal predicted the location of a reward, measured as the distance of the first lick in a given trial to the start of the reward zone (Figure 4—figure supplement 1C,D, subsection “Trial onset, landmark, and reward encoding neurons in RSC”). Population analyses further showed a significantly more accurate representation of space in expert compared to naïve animals (Figure 6C-E). These results provide evidence for both single neuron and population-based codes for landmarks and rewards in RSC.

9) Regarding laminar differences in landmark coding, the justification is unclear for this. There is no rationale as to what differences would be expected in terms of the connectivity of the different layers. The authors write "….L2/3 may be key for encoding landmarks during ongoing behaviour". One may assume this is because there is a greater proportion of cells engaged, but greater overall numbers does not necessarily a greater role – it may be more informative to model how networks within RSC support behavior i.e., interactions between the layers.

The role(s) of different layers in any cortex is poorly understood. We thus had no strong conceptual framework upon which to base potential hypotheses for our L2/3 vs. L5 RSC imaging. We therefore deliberately present this data as an observation rather than an answer to a specific question. Very few studies have imaged from multiple layers in behaving animals anywhere in the cortex (Funamizu, Kuhn, and Doya, 2016; Mao, Kandler, McNaughton, and Bonin, 2017), making our data a useful resource for the field. Additionally, despite existing electrical recording data that originates from multiple layers, there are few experimental or theoretical studies that address the difference(s) in coding between L2/3 vs. L5. This makes it virtually impossible to formulate meaningful predictions about what functional differences we might expect between layers, particularly in an association cortex like RSC, during a complex task. Furthermore, to the best of our knowledge, there is no existing framework for modeling how interactions among layers in any cortex support behavior. We believe that attempting to build an interacting multi-layer cortical model that explains behavior is outside the scope of this study. Additionally, our knowledge of RSC is still quite limited (which was the motivation for our study) and thus any model we could develop at this point would likely still not be appropriately constrained. The figure on laminar differences was removed from the main text and added as a supplemental figure due to space constraints. A brief section describing the findings has been added to the main text (paragraph three subsection “Trial onset, landmark, and reward encoding neurons in RSC”).

10) The present study appears to be focused on dysgranular RSC; the main RSC subregions are known to have different connectivity and responsiveness, so the authors should be more specific in terms of which part of RSC they are referring to. Moreover, the authors distinguished different RSC layers but did not evaluate possible differences between granular and dysgranular parts of the RSC. Given that these two areas contain functionally different head-direction cell categories (Jacob et al., 2017), is there any possibilities that they respond differently to landmarks? This should be clarified.

We acknowledge the important distinction between granular (A29) and dysgranular (A30) portions of RSC. All our recordings were carried out in dysgranular (A30) RSC. We have adapted language throughout the manuscript to reflect this.